# Surface melt and the importance of water flow – an analysis based on high-resolution UAV data for an Arctic glacier

Eleanor A. Bash[1] and Brian J. Moorman[1]

[1]Department of Geography, University of Calgary, Calgary, Alberta, Canada

**Correspondence:** Eleanor A. Bash (eleanor.bash@gmail.com)

**Abstract.** Models of glacier surface melt are commonly used in studies of glacier mass balance and runoff, however, with limited data available most models are validated based on ablation stakes and data from automatic weather stations (AWS). The technological advances of unmanned aerial vehicles (UAVs) and structure-from-motion (SfM), have made it possible to measure glacier surface melt in detail over larger portions of a glacier. In this study, we use melt measured using SfM processing of UAV imagery to assess the performance of an energy balance (EB) and enhanced temperature index (ETI) melt model in two-dimensions. Imagery collected over a portion of the ablation zone of Fountain Glacier, Nunavut, on July 21, 23, and 24, 2016 was previously used to determine distributed surface melt. An AWS on the glacier provides some measured inputs for both models, as well as an additional check on model performance. Modelled incoming solar radiation and albedo derived from UAV imagery, are also used as inputs for both models which were used to estimate melt from July 21-24, 2016. Both models estimate total melt at the AWS within 16% of observations (4% for ETI). Across the study area the median model error, calculated as the difference between modelled and measured melt (EB=-0.064 m, ETI=-0.050 m), is within the uncertainty of the measurements. The errors in both models were strongly correlated to the density of water flow features on the glacier surface. The relation between water flow and model error suggests that energy from surface water flow is contributing significantly to surface melt on Fountain Glacier. Deep surface streams with highly asymmetrical banks are observed on Fountain Glacier, but the processes leading to their formation are missing in the model assessed here. The failure of the model to capture flow-induced melt would lead to significant underestimation of surface melt should the model be used to project future change.

## 1 Introduction

The Canadian Arctic Archipelago holds approximately 14% of the world's area of glacier ice outside the major ice sheets and rates of glacier melt in the region have increased since the late 1990s (Gardner et al., 2011; Noël et al., 2018; Lenaerts et al., 2013; Gardner et al., 2012). Fisher et al. (2012) show that recent melt rates on Canadian Arctic ice caps are the highest in 4000 years, while Gardner et al. (2012) found that glaciers of the southern Canadian Arctic Archipelago contributed over 16% of 2003-2010 sea-level rise. Future projections indicate the glaciers from across Arctic Canada will continue to be the largest

mountain glacier contributors to sea-level rise through the end of this century (Radić and Hock, 2011). Given the importance of Canadian Arctic glaciers to global sea-level rise and the rapid change in melt rates observed in recent years, it is critical to better understand the processes contributing to melt rates on these glaciers.

Direct measurements of melt rates on Arctic glaciers are scarce, only five glaciers from the Canadian Arctic have current data available online (WGMS, 2018). Where in situ data is collected, it is often based on ablation stake networks and data from automatic weather stations (AWS; e.g. Bash et al., 2018; WGMS, 2018). These in situ measurements must be extrapolated to provide estimates of melt at other locations on a glacier, or at other glaciers where no measurements exist. Several kinds of models are commonly used to extrapolate glacier surface melt measurements, including temperature index (TI; e.g. Hock, 1999, 2005), enhanced temperature index (ETI; e.g. Irvine-Fynn et al., 2014; Bash and Marshall, 2014), and energy balance (EB; e.g. MacDougall and Flowers, 2011; Shaw et al., 2016). The TI or ETI models are often preferred for regional scale models, because of their lower computational needs and the ease with which input variables can be estimated. Energy balance models require more inputs and as a result are preferred for in depth studies of individual glaciers.

Lack of data for Arctic glaciers also makes it difficult to validate model results. Datasets are often split into training and validation periods to assess model performance, however, these measurements represent only a few locations on the glacier surface (Mair et al., 2003; Wake and Marshall, 2015; Matthews et al., 2015). Where outlet streams are measured, studies use total stream discharge and aggregated surface melt to validate results (e.g. Pellicciotti et al., 2005; Bash and Marshall, 2014). Validation with total melt provides insight into the average performance of these models over an entire glacier, but neither this method or the validation based on point data provide insight into model performance across a glacier surface.

Technological advances have allowed for increasingly detailed change detection from imagery obtained by satellites, unmanned aerial vehicles (UAVs), and terrestrial photography. Structure-from-motion (SfM) is widely used across a range of disciplines for reconstructing topography using imagery from the aforementioned sources (e.g. James and Robson, 2014; Cook, 2017; Watson et al., 2017; Lovitt et al., 2018; Bash et al., 2018). Employing SfM, Watson et al. (2017) use multiple reconstructions of a debris covered glacier to measure seasonal ice cliff retreat from terrestrial photographs, including spatial variation in rates across individual cliff faces. Using similar methods and UAV imagery, Bash et al. (2018) measured spatially variable melt rates in the ablation zone of an Arctic glacier over 4 days. This new capacity for measuring change in high temporal and spatial detail provides opportunities to examine spatial patterns in ways that were previously not possible.

The methods of surveying that are needed for detailed surface change measurements using SfM are time consuming (Watson et al., 2017; Bash et al., 2018; Avanzi et al., 2018). These surveys involve multiple site visits, measurement of control points using differential GPS or total stations, and imagery collection (Bash et al., 2018). Given the effort involved in these data collection campaigns, it is not feasible at this time to extend these methods to regional-scale glacier change detection. Modelling efforts will continue to play an important role in understanding glacier melt in the future and thus it is critical to improve the models employed.

The aim of this study is to use melt measurements available in high spatial and temporal resolution to assess the performance of an energy balance model and an enhanced temperature index model across the glacier surface. We do this using previously

published surface melt data which was measured using UAV surveys in the ablation zone of Fountain Glacier, Nunavut (Bash et al., 2018).

## 2 Methods

### 2.1 Study Area

Fountain Glacier, located on Bylot Island, Nunavut (Figure 1), has been studied in detail since 1991 (e.g. Moorman, 2003; Whitehead et al., 2014; Bash et al., 2018). The glacier stretches 16 km from higher elevations in the Byam Martin Mountain Range to its terminus roughly 10 km inland from the coast. The focus area of this study, in the lower ablation zone, spans a narrow elevation range (250–400 m) and terminates in a cliff face with two calving fronts. The lower portion of the glacier faces east, with several supraglacial streams of varying sizes (St. Germain and Moorman, 2016). The largest of these streams form deeply incised asymmetrical canyons, with a steeper north facing valley wall and a more gently sloping south facing valley wall (Figure 1).

During July of 2016 aerial surveys were conducted with a UAV to reconstruct 0.185 km$^2$ of the glacier surface multiple times (Bash et al., 2018). Bash et al. (2018) measured surface lowering between July 21 and 24 using point cloud differencing across the study area indicated in Figure 1. Concurrently, an automatic weather station (AWS; Figure 1) recorded surface melt with a Campbell Scientific SR50 sonic ranger, as well as temperature, incoming and outgoing shortwave radiation, relative humidity, wind speed and direction. Bash et al. (2018) assessed the surface lowering measured through point cloud differencing against 17 ablation stakes, as well as the AWS data, and found that the measured surface lowering agreed with other melt measurements throughout the study area. In this study we assume that point cloud derived surface lowering is equivalent to surface melt. The uncertainty of this surface lowering was 0.048 m.

Bash et al. (2018) measured an average daily melt rate of 0.060 m day$^{-1}$ across the study area between July 21-24, 2016. Between 2010–2011, Whitehead et al. (2014) measured melt rates of 0.030–0.055 m day$^{-1}$ across the ablation zone. Bash et al. (2018) also showed, however, that melt was highly variable across the study area during July 2016, ranging from 0.010–0.110 m day$^{-1}$. The authors found that these differences in melt were not tied to elevation or aspect of the glacier surface.

### 2.2 Description of Data

All AWS measurements were taken at 2 min intervals and recorded as hourly averages from July 1 to July 31, 2016 (Figure 2). The SR50 was installed on a pole drilled into the glacier surface immediately next to the AWS on July 13 and measured hourly average surface height. The pole holding the SR50 began to tilt due to melt out of the pole during the afternoon of July 27 and was reinstalled in the morning on July 28. The data from that time period was removed from the time series for model validation.

Models run at high temporal resolution can have significant errors which stem from SR50 readings (Matthews et al., 2015). These errors stem from the uncertainty in SR50 readings due to the instrument accuracy (0.01 m), as well as uneven topography

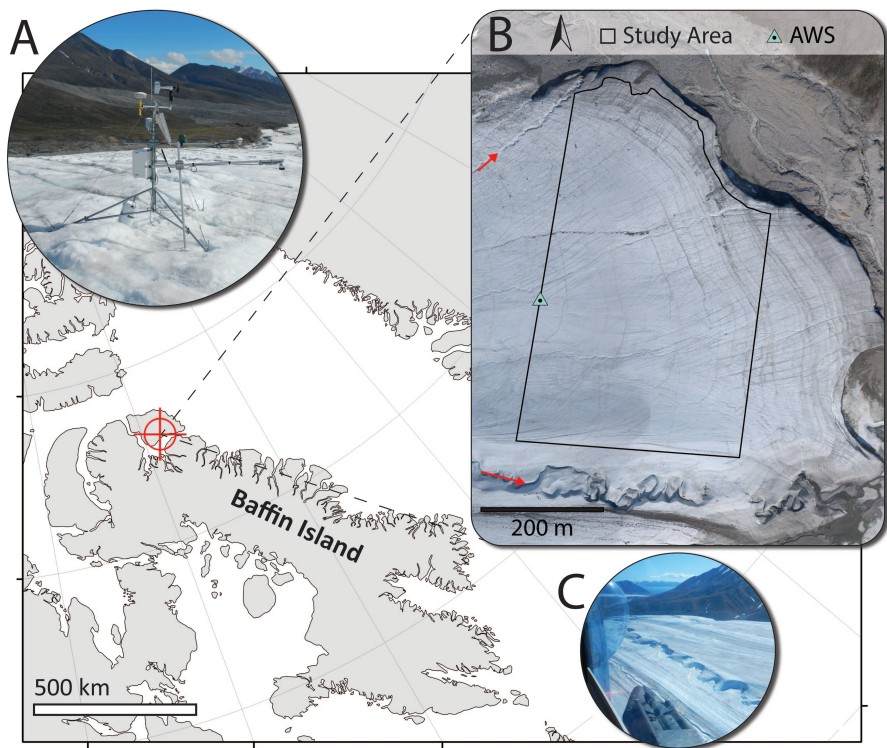

**Figure 1.** Location of Fountain Glacier on Bylot Island, Nunavut, in the Canadian Arctic. (**A**) The automatic weather station (AWS) installed on a tripod, measuring incoming and outgoing shortwave radiation and temperature. An SR50 measuring surface position was installed on a separate pole drilled into the glacier (foreground). (**B**) A 2011 orthomosaic of the lower ablation zone of Fountain Glacier. The boundary of the study area is shown, the area extends to the 2016 glacier boundary on the north side. The AWS location is also indicated (green triangle). Two canyons formed by stream incision are indicated by red arrows. (**C**) An aerial view of the largest incised canyon on Fountain Glacier. The photo is taken facing east and the steep north facing canyon walls can be seen in contrast to the more gently sloping south facing walls.

underneath the instrument. In addition to hourly surface height, the SR50 records a standard deviation of all measurements in the hourly average (typically 30 measurements). SR50 values where the standard deviation was greater than 0.01 m were removed, expected hourly melt rates are lower than the instrument accuracy and standard deviations greater than that are assumed to be related to noise. After removing those points, a 5 hr rolling average was calculated to smooth the data and fill in removed values. The 5 hr window was used to assure that all windows had multiple measurements to average, after some measurements had been removed. SR50 values which differed significantly from others in the 5 hr window were also removed (defined as more than the standard deviation of hourly change in the 5 hr window), under an assumption that it was physically unlikely for melt rates to change dramatically over short periods. Melt was then calculated in 12 hr periods (0:00 – 12:00) using the remaining values.

Processing of UAV imagery was done in Agisoft Photoscan and is described in detail by Bash et al. (2018). High accuracy settings were used to retain detail in the final point cloud. During processing, point clouds of the glacier surface at mid-day were produced for July 21 and 24, 2016. The point clouds were used to create orthomosaics and digital surface models (DSMs) for each day with horizontal resolutions of 0.10 m.

The multiscale model to model cloud comparison algorithm (M3C2) was used to calculate melt across the study area at 0.02 m horizontal resolution (Lague et al., 2013; Bash et al., 2018), with vertical uncertainty of 0.048 m based on the RMSE of surface lowering compared to ablation stakes. The M3C2 algorithm calculates change normal to the local surface, which was desirable for measuring detailed change at such fine resolution. After differencing the point clouds were filtered to remove large outliers and differences calculated from fewer than 5 points, as suggested by Lague et al. (2013). This filtering resulted 10  in variable density of points in the final point cloud. To align with DSMs and orthomsaics of the study area, for this study melt data was gridded to 0.10 m resolution. Melt measurements were averaged within each grid cell and empty cells were filled using bilinear interpolation.

## 2.3    Model Formulation

### 2.3.1    Energy Balance Model

An EB model accounts for all energy inputs and outputs at the glacier surface to estimate energy available for melt or freezing. As a representation of surface energy exchange, these models are often assumed to be the best estimators of surface melt (Hock, 2005). We employ an energy balance model described by Ebrahimi and Marshall (2016) as a check on both SfM-derived change and the ETI model described below.

$$Q_N = Q_{SW}^{\downarrow} - Q_{SW}^{\uparrow} + Q_L^{\downarrow} - Q_L^{\uparrow} + Q_H + Q_E \tag{1}$$

where $Q_N$ is the net energy flux at the surface and $Q_{SW}^{\downarrow}, Q_{SW}^{\uparrow}, Q_L^{\downarrow}, Q_L^{\uparrow}, Q_H, Q_E$ represent incoming and outgoing shortwave radiation, incoming and outgoing longwave radiation, sensible and latent heat flux, respectively. The energy fluxes are all calculated in MJ m$^{-2}$ hr$^{-1}$ for consistency with AWS measurements and energy fluxes are positive when they contribute energy to the glacier surface. When $Q_N > 0$ melt ($\hat{M}$; m ice) is calculated by:

$$\hat{M} = \frac{Q_N}{\rho_{ice} L_f} \tag{2}$$

where $\rho_{ice}$ is the density of glacier ice, assumed to be 900 kg m$^3$ (e.g. Arnold et al., 2006; Cuffey and Paterson, 2010; Fitzpatrick et al., 2017) and $L_f$ is the latent heat of fusion. Longwave radiation ($Q_L$) is parameterized based on the absolute temperature (T) and emissivity ($\varepsilon$):

$$Q_L = \varepsilon \delta T^4 \tag{3}$$

where $\delta$ is the Stefan-Boltzmann constant. For $Q_L^\uparrow$, the calculation is straightforward if the surface is assumed to be melting (273.15 K), the surface emissivity is then 1.0. The atmospheric emissivity ($\varepsilon_a$) must be estimated in order to obtain $Q_L^\downarrow$. Here we use a parameterization from Ebrahimi and Marshall (2015) based on vapour pressure ($e_v$), and relative humidity ($H$), which was found to be robust when transferred directly without local training.

$$\varepsilon_a = 0.445 + 0.0055H + 0.0052e_v \tag{4}$$

The sensible and latent heat fluxes are calculated by equations 5 and 7, respectively.

$$Q_H = \rho_a c_p k^2 v \left[ \frac{T_{pa} - T_{ps}}{ln(z/z_0)ln(z/z_{0H})} \right] \tag{5}$$

where $\rho_a$ is the air density, calculated from the near surface air temperature and average air pressure at the AWS ($P_{aws}$=969), $c_p$ is the heat capacity of air, $v$ is the measured wind speed, and $k = 0.4$ is von Karman's constant. The potential temperature is calculated for the near surface air ($T_{pa}$) and ice surface by($T_{ps}$):

$$T_p = T \frac{P_{ref}}{P_{aws}}^{R/c_p} \tag{6}$$

where $P_{ref}$ is a reference pressure (1000 mb), $R = 288.5$ and the surface is assumed to be at the melting point.

$$Q_E = \rho_a L_v k^2 v \left[ \frac{q_a - q_s}{ln(z/z_0)ln(z/z_{0E})} \right] \tag{7}$$

where $L_v$ is the latent heat of vaporization, $q_a$ is the specific humidity, based on T and $H$, and $q_s$ is the specific humidity calculated from the saturation vapor pressure at the melting surface. $z_0$, $z_{0H}$, and $z_{0E}$ are the roughness length scales for turbulent exchange of momentum, heat, and moisture. The height of all AWS measurements is $z = 1.5$ m and $z_0 = 0.005$ m, was used as a tuning parameter at the AWS. Both equation 5 and 7 neglect a correction for atmospheric stability, which has been shown in previous studies to provide better results for these parameters (Hock and Holmgren, 2005; Ebrahimi and Marshall, 2016)

A 1-D formulation of this model (EB$_{aws}$) runs using measured inputs from the AWS. A distributed version was also developed (EB$_{dist}$) using modelled incoming radiation at the AWS location and albedo derived from the UAV imagery to determine SW$_{net}$ (both described below, Figure 2A, B, 3). Near surface air temperature across the study area was modelled using the uncorrelated AWS temperature and modelled radiation (Section 2.3.2). Measured values from the AWS are used for all other variables in equations 3 - 7.

Distributed incoming radiation was modelled by modifying the hourly measured incoming radiation at the AWS by the slope ($S$) and aspect ($A$) of each grid cell in the July 21 DSM (Figure 3C). The terrain correction was based on Goswami et al. (2000).

$$SW_{in} = Q_{SW}^\downarrow * cos\psi * cos(\omega - A) * sinS * cosS \tag{8}$$

where $\psi$ is the solar altitude and $\omega$ the solar azimuth. This correction alone causes radiation to drop to zero overnight, which is not realistic during Arctic summers. Although direct radiation may drop to zero at some cells when sun angles are low overnight (solar elevation dips to 3.6°), diffuse radiation is at measurable levels (Figure 2E). A diffuse correction ($SW_{diff}$; based on Bugler (1977)) was added to incoming radiation after correction for incidence angle to estimate the total incoming radiation (Figure 2F).

$$SW_{diff} = 16\psi^{0.5} - 0.4\psi \tag{9}$$

Lower peak radiation in the model (Figure 2B) is due to a slight difference in slope and aspect between the radiometer (which was level) and the grid cell of the DSM containing the AWS (slope = 5°, aspect = 177°). Estimated radiation at each cell was then multiplied by $(1 - \alpha)$ to estimate $SW_{net}$ at each cell.

Albedo was estimated across the study area using the orthomosaics from July 21 and July 23. Each orthomosaic contains the digital number recorded by the camera in the red, green, and blue (RGB) channels. The digital number cannot be directly used to calculate surface reflectance without conversion equations proprietary to the camera manufacturer. However, several studies have used other approaches to derive surface reflectance from digital numbers (Corripio, 2004; Rippin et al., 2015; Ryan et al., 2017). Rippin et al. (2015) use the sum of RGB digital number values as a proxy for albedo, similar to the approach we employ here.

Values for RGB channels were averaged and the total range scaled to match the value at the AWS location (Figure 3). Scaling in this way allows the values to be directly input into models, as opposed to providing a proxy for albedo as was done in Rippin et al. (2015). The average measured albedo value for the study period (0.31; July 21-24, 2016) was used to fix the rescaled value for both July 21 and July 23 at the AWS location. The lower end of the range was based on albedo values of cryoconite holes reported by Ryan et al. (0.1; 2017), an assumption was made that cryoconite holes would have similar albedo to debris on the glacier surface. The scaling of RGB values was used to create two gridded albedo products that could be directly input into the melt model (Figure 3). July 21 albedo was used as a model input up to 12:00 July 23, while the July 23 albedo was used from 12:00 July 23 onward.

Mean albedo over the grid was 0.35 for July 21 and 0.33 for July 23, the median difference between the two was 0.020 (Figure 3). Albedo measured at the AWS on July 21 and July 23 at 12:00 (the time of image acquisition) was 0.315 and 0.309. The more pronounced difference in the gridded albedo likely reflects darker imagery in the July 23 mosaic, in addition to a true lowering of the surface albedo over the time period. To test the importance of this lower albedo value, total absorbed radiation at the AWS for July 2016 was adjusted to reflect a lower albedo, beyond what was measured at the AWS. This lowering of 0.014 could produce approximately 0.028 m additional melt over the course of the month.

### 2.3.2 Enhanced Temperature Index Model

This study assesses the performance of an enhanced temperature index model because these models are commonly used for regional scale estimations of glacier melt. The model we use is formulated after Bash and Marshall (2014), which uses absorbed radiation ($SW_{net}$; MJ hr$^{-1}$) and temperature (°C) in a linear regression model to estimate melt ($\hat{M}$; m). The formulation of the

**Table 1.** Model coefficients obtained using linear regression modelling for July 14 – 20, 2016. Units of $TF$ are $\mathrm{mh^{-1}{}^{\circ}C^{-1}}$, and $RF$ are $\mathrm{mh^{-1}MJ^{-1}}$

|  | $TF$ | $RF$ |
|---|---|---|
| This study | 0.00019 | 0.0015 |
| Irvine-Fynn et al. (2014) maximum | 0.00068 | 0.0017 |
| Irvine-Fynn et al. (2014) minimum | 0.0002 | -0.0014 |
| Pellicciotti et al. (2005) | 0.00005 | 0.0026 |

model is unique in that it controls for correlation between independent variables which can make model results unstable when not addressed. The uncorrelated temperature ($\mathrm{T}_{residual}$) is calculated by:

$$T_{SW} = \eta + \beta SW_{net} \tag{10}$$

$$T_{residual} = T - T_{SW} \tag{11}$$

where $\eta$ and $\beta$ are coefficients fit using a linear regression. $\mathrm{T}_{residual}$ is then used in the following equation to determine melt in meters of surface lowering consistent with SR50 and UAV measurements:

$$\hat{M} = \begin{cases} TF \cdot T_{residual} + RF \cdot SW_{net} & : T > T_T \\ 0 & : T \leq T_T \end{cases} \tag{12}$$

where $\mathrm{SW}_{net}$ can be calculated by the difference between incoming and outgoing radiation, or by multiplying incoming radiation by $(1 - \alpha)$, where $\alpha$ is the surface albedo. The coefficients $TF$ and $RF$ were fit using a linear regression. Threshold
temperatures ($T_T$) ranging from $1^{\circ}$- $2^{\circ}$C have been used in previous studies (Hock, 1999; Pellicciotti et al., 2005; Gabbi et al., 2014; Irvine-Fynn et al., 2014; Matthews et al., 2015). Here we use $T_T = 0^{\circ}$C. Although we have not performed any testing of this threshold, it is unlikely that it would affect model outcomes in this study as the air temperature rarely fell below $2^{\circ}$C during the month of July 2016, and never during the period where distributed melt information is available.

Cumulative positive $\mathrm{T}_{residual}$ and $\mathrm{SW}_{net}$ were calculated for 12 hr periods beginning July 14 at 12:00. Data from July 14 –
July 20 was used as a training period to determine the model coefficients ($\eta$, $\beta$, TF, RF) using a multiple linear regression. The fitted model coefficients are shown in Table 1. Model coefficients compare well to those reported by Irvine-Fynn et al. (2014), a study which also modelled an Arctic glacier, although they report a wide range of values for $RF$ depending on the training period.

A 1-D formulation of the model was run at the AWS site ($\mathrm{ETI}_{aws}$) using measured $\mathrm{SW}_{net}$ and near surface air temperature.
A distributed model ($\mathrm{ETI}_{dist}$) was developed by estimating the model variables across the study area at each grid cell. Given the

relatively small size of the study area, $T_{residual}$ was assumed to remain constant across the area, calculated by equations 10 and 11 with modelled incoming radiation and measured temperature at the AWS site. Equations 10 and 11 can be used to back-calculate temperature when $T_{residual}$ and $SW_{net}$ are known, this allowed modelled temperature to vary across the grid. $SW_{net}$ was calculated from modelled distributed radiation and albedo, described above.

## 2.4 Model Performance

The four models ($EB_{aws}$, $ETI_{aws}$, $EB_{dist}$, $ETI_{dist}$) were run for the period 0:00 July 1–24:00 July 31, 2016, allowing for comparison between models over a longer time period than when melt measurements were available. Outputs from 0:00 July 14–24:00 July 31 were compared directly to melt measured with the SR50. To compare $EB_{dist}$ and $ETI_{dist}$ estimates, melt values were extracted from the grid cell containing the AWS. All models were assessed using the mean and standard deviation ($\sigma$) of the residuals ($\hat{M} - M$).

Total melt estimates for July 21–24 from both distributed models were compared to the gridded melt measurements across the study area for the same period. Model error was calculated at each grid cell in the study area, negative values indicate underestimation and positive values indicate overestimation of the model as compared to the UAV-derived data ($\hat{M} - M$). The median model error (ME) and normalized median absolute deviation (NMAD) were used to describe the model error following Höhle and Höhle (2009). Höhle and Höhle (2009) suggest using these metrics to describe errors when the error distributions are non-normal, which is typical for the large datasets produced with SfM (Montgomery and Runger, 2007). In the case of non-normal distributions, standard descriptive statistics (based on data means and standard deviations) are disproportionately influenced by outliers, which the ME and NMAD are not (Maronna et al., 2006). The median and range of measured and modelled melt were also calculated for comparison.

In addition to model performance, the model error across the study area was used to investigate factors influencing model performance. We examined model errors in relationship to surface characteristics which are known to influence the energy available for melt (Hock, 2005). The Pearson correlation coefficient is often used to assess variable relationships:

$$R = \frac{\sum_{i=1}^{n}(x_i - \bar{x})(y_i - \bar{y})}{\sigma_x \sigma_y} \tag{13}$$

where $\bar{x}$ and $\bar{y}$ are the variable means, $\sigma_x$ and $\sigma_y$ are the variable standard deviations. As noted above however, in the case that the mean and standard deviation are poor descriptors of a population, an alternative measure of relationship is necessary using robust estimators for the location and variance (Shevlyakov and Smirnov, 2011). Based on Shevlyakov and Smirnov (2011), we use the median correlation coefficient to assess the relationship between model error and potential explanatory variables.

$$R_r = \frac{\sum_{i=1}^{n}\frac{1}{n}(x_i - med(x))(y_i - med(y))}{NMAD_x NMAD_y} \tag{14}$$

In the case of large samples such as we have here (n>18 million), p-values are not a suitable measure for correlation significance (Maronna et al., 2006), instead we evaluate the strength of the correlation based on its value. Robust correlations were computed between model error and terrain variables (slope and aspect), albedo, and an estimate of surface water flow.

Water flow direction and potential upstream catchment were quantified using the Hydrology Toolset in ArcGIS 10. Assuming that water is produced at every grid cell, flow paths were calculated from the July 21 DSM based on potential upstream accumulation. Cells with more than 1500 upstream cells were converted to a set of linear features and the number of these features within 20 m of each grid cell was calculated ($W_{20m}$). The 1500 cell threshold represents 15 m$^2$ of upstream drainage area, and was chosen based on the assumption that 15 m$^2$ is likely to provide enough melt water to start flow. The resulting layer provided a representation of areas with significant surface water flow, including streams and thin sheet flows.

## 3 Results

### 3.1 Inter-model comparison at AWS

The four models were compared at the AWS site for July 1-31, 2016. All four models capture daily patterns of melt, with lower melt totals on average in the second half of the month, which is consistent with lower solar radiation recorded at the AWS (Figures 4A and 2B). The distributed models both consistently estimate lower melt totals than their point model counterparts, with differences more pronounced when melt values are high. This is most likely related to the lower shortwave radiation in the distributed model, which is in total 17% less than measured for the study period (Figure 2B). The EB models produce lower melt estimates than the ETI models when solar radiation is low. During periods of high incoming solar radiation, $EB_{aws}$ and $ETI_{aws}$ produce similar melt.

The total melt estimated during the month of July was lower in the two distributed models, $EB_{dist} = 1.223$ m and $ETI_{dist} = 1.394$ m, compared to the point models, $EB_{aws} = 1.436$ m and $ETI_{aws} = 1.508$ m. The EB models estimate less melt for the month than the ETI counterparts.

### 3.2 Model performance at AWS site

Model estimates from July 14-31, 2016 were compared to melt measured at the AWS site with the SR50. Total melt at the SR50 during the study period was $0.186\pm0.03$ m, while the imagery-derived measurement for the same period was $0.274\pm0.048$ m. After tuning $z_0$, the mean residual of 12-hr model estimates at the AWS was 0.000 m for $EB_{aws}$ and -0.003 m for $EB_{dist}$. The fitted $ETI_{aws}$ also had a mean residual of 0.000 m, while that of $ETI_{dist}$ was -0.001 m. Three of the four models underestimate melt at the site, $ETI_{aws}$ over estimates melt by 0.016 m (Table 2; Figure 4B). Both distributed models produce lower melt estimates than their point model counterparts. The largest underestimate was from $EB_{dist}$. The variability of modelled melt is notably lower than measured melt in all cases.

**Table 2.** Descriptive statistics for measured melt, modelled melt, and model residuals at the AWS site, July 14-31, 2016. All values were calculated based on 12-hr melt estimates and are reported in meters. We denote the standard deviation of all 12-hr melt values with $\sigma$ and the standard deviation of all residuals in 12-hr model estimates with $\sigma_{Res}$.

|  | Measured | $EB_{aws}$ | $ETI_{aws}$ | $EB_{dist}$ | $ETI_{dist}$ |
|---|---|---|---|---|---|
| Mean | 0.022 | 0.022 | 0.023 | 0.019 | 0.021 |
| $\sigma$ | 0.010 | 0.008 | 0.006 | 0.006 | 0.006 |
| Total Residual |  | -0.030 | -0.126 | 0.016 | -0.029 |
| Mean Residual |  | 0.000 | 0.000 | -0.003 | -0.001 |
| $\sigma_{Res}$ |  | 0.009 | 0.011 | 0.009 | 0.010 |

## 3.3 Model performance over the study area

Across the study area median measured melt was 0.184±0.048 m, the NMAD was 0.026 m (Figure 5B). Median melt across the area was lower in both $EB_{dist}$ and $ETI_{dist}$, as was the variability (Figure 5A,C). The total range of measured melt was 0.305 m, while the range of $EB_{dist} = 0.123$ and $ETI_{dist} = 0.083$.

5      The ME and NMAD of $EB_{dist} = -0.064 \pm 0.022$, while that of $ETI_{dist} = -0.050 \pm 0.023$. The ME of $ETI_{dist}$ is similar magnitude to the measurement uncertainty (0.048 m), indicating general agreement with measurements. When areas of error above the measurement uncertainty are highlighted (Figure 5D,E), $ETI_{dist}$ shows clustered patterns representing 53% of the total study area. Errors greater than twice the uncertainty (0.096 m) account for 5% of the study area. $EB_{dist}$ errors greater than 0.048 m cover 79% of the study area, while those greater than 0.096 m cover 11%.

10      Model uncertainty can be derived from a number of metrics, including variability in the model residuals, represented by the NMAD or standard deviation. Based on assessment at the AWS site, uncertainty in 12-hr estimates from $EB_{dist}$ and $ETI_{dist}$ are $2\sigma_{Res}$ = 0.018 m and 0.022 m, respectively. Bash and Marshall (2014) use validation at an AWS and incorporate uncertainties related to modelling input variables to estimate an uncertainty of 15%. In this study, we also have the benefit of examining distributed results across the study area through comparison with UAV derived melt. The variability of the distributed

15 models is characterized by the NMAD, taking $2*NMAD = 0.028$ m for $EB_{dist}$ and $2*NMAD = 0.044$ m for $ETI_{dist}$, these uncertainties are 4-6% of total estimated melt. Combined with the uncertainty in the measured surface change (0.048 m), a total uncertainty in the model errors can be derived ($\sqrt{2*NMAD^2 + 0.048^2}$), $EB_{dist} = 0.056$ m, $ETI_{dist} = 0.065$. In either case, the errors which are strongly correlated with $W_{20m}$ are higher than the combined uncertainty of measurement and model, particularly for $ETI_{dist}$.

## 20    3.4 Relationship to surface characteristics

Moderate correlation was found between model error and albedo, aspect, and slope for both $EB_{dist}$ and $ETI_{dist}$ (Table 3). Through field observation, we were able to note a visual relationship between surface water features and high error during

**Table 3.** Robust correlation statistics ($R_r$) for surface characteristics and distributed models. Strong correlations are highlighted in bold.

|        | $EB_{dist}$ | $ETI_{dist}$ |
|--------|-------------|--------------|
| Albedo | -0.207      | -0.208       |
| Aspect | 0.177       | 0.178        |
| Slope  | 0.204       | 0.197        |
| $W_{20m}$ | **0.470** | **0.462**    |

analysis (Figure 6A, C). This association was confirmed through correlation with $W_{20m}$, $R_r = 0.470$ for $EB_{dist}$ and $R_r = 0.462$ for $ETI_{dist}$.

Although most of the errors are negative, positive errors (i.e. overestimation) were also found. These are primarily associated with boulders and other large stationary objects on the glacier surface.

## 4   Discussion

Model results at the AWS indicate that both tuned point models perform very well. Performance of the distributed models is poorer, with $EB_{dist}$ performing worse than $ETI_{dist}$, despite the physical basis of the energy balance model. This is likely due to a combination of factors, including limited data inputs available, and need for further refinement of energy exchange parameterizations. Lower modelled melt across the study area can be partly explained by lower modelled radiation, which was noted at the AWS site. However, the discrepancy between measured and modelled melt is greater across the grid than at the AWS site, indicating the difference is not due to solar radiation alone. Previous work has shown that energy balance model performance is sensitive to variations in roughness length, wind speed, and albedo (Hock and Holmgren, 2005; MacDougall and Flowers, 2011). Of these components, roughness length is the only EB model input used here that is not measured. An analysis of the model sensitivity to $z_0$ revealed a 10% decrease in total estimated melt (July 1-31) for $z_0 = 0.001$ m, compared to the reference run with the tuned $z_0$. Similarly, $z_0 = 0.01$ m resulted in a 7% increase in estimated melt. The deviation from the reference model run was caused by a 33% decrease and 22% increase in both $Q_H$ and $Q_E$. Despite this sensitivity, $z_0 = 0.005$ provided the best fit for the training data.

Variability in both EB and ETI models is linked to solar radiation cycles. In addition to diurnal cycles in $SW_{in}$ due to zenith angle variations, topography blocks direct solar radiation for a time during the night and causes stronger diurnal variation in $SW_{in}$. The strength of the signal from the diurnal cycle is much greater in the melt produced by EB models than the ETI models (Figure 4A). As noted above, on clear sky days with relatively low temperatures, the EB and ETI models produce similar results (July 1-3). Under the opposite conditions, low $SW_{in}$ and high temperatures (e.g. July 14-15, 28-29), the EB and ETI models produce the most dissimilar results. This is due to the relationship between temperature and radiation in the ETI model, where at low temperatures and high $SW_{in}$, $T_{residual}$ is very small and melt is driven primarily by radiation. At high temperatures and low $SW_{in}$, $T_{residual}$ is relatively larger and drives melt. In contrast, melt in the EB model is always primarily

driven by solar radiation. Examining 12-hr averages of $SW_{net}$, $LW_{net}$, and $Q_h + Q_E$ in the EB model outputs, there is a strong correlation between $SW_{net}$ and available melt energy ($R = 0.90$). The correlation between melt energy and $LW_{net}$, as well as $Q_h + Q_E$, is much lower ($R = 0.25$ and $R = -0.19$, respectively).

This contrast between the EB and ETI models, and the local tuning of the ETI model, allows $ETI_{dist}$ to better estimate melt recorded at the AWS. For comparison to other studies, $\sigma_{Res}$ from $ETI_{dist}$ was converted to an hourly rate ($ETI_{dist} = 0.0005$ m h$^{-1}$), which is similar to other ETI models in the literature (Pellicciotti et al., 2005; Irvine-Fynn et al., 2014; Bash and Marshall, 2014). Irvine-Fynn et al. (2014) report a standard error of $0.00018 - 0.00053$ m h$^{-1}$, the range in error stems from model runs with different training datasets. Hourly ETI model error reported by Pellicciotti et al. (2005) is as high as 0.0055 m water equivalent h$^{-1}$, compared to an energy balance model as reference data. Bash and Marshall (2014) report a standard error of 0.00031 m h$^{-1}$, when the ETI model is transferred in time. Overall these results at the AWS indicate that, in the absence of additional validation data, it would be reasonable to employ this model to estimate melt across the glacier surface.

Hock (1999) found that when TF is calculated based on melt and temperature records, it is highly variable. The inclusion of shortwave radiation in ETI models is an attempt to account for this variability. The low variability seen in the results of both $ETI_{aws}$ and $ETI_{dist}$ when compared to AWS measurements is an indication of potential problems in the ETI models. Similar variability was noted in other studies (Hock, 1999; Bash and Marshall, 2014). However, Hock (1999), Pellicciotti et al. (2005), and Irvine-Fynn et al. (2014), all cite $R^2$ values of 0.60, 0.86, and 0.80 when comparing total modelled melt to measured discharge. These previous results show that although the ETI models capture total melt, melt rates at individual points are captured less effectively. Similarly, we found across the study area that median modelled and measured melt agreed, but model results showed variability an order of magnitude lower than measurements (Figure 5B, C). The availability of melt measurements at 10 cm horizontal resolution shed light on the areas where modelled and measured melt were (or were not) in agreement, which has not been shown before. These measurements also allowed us to examine possible explanations for the model errors, including the aspect and surface water flow.

It is worth noting that surface lowering measured through point cloud differencing is not always equivalent to surface melt. Vertical ice motion in the ablation zone offsets melt in measurements of surface change. However, in the case of Fountain Glacier Whitehead et al. (2014) estimate surface uplift to be 0.2 m over the course of the ablation season, or 0.002 m d$^{-1}$. Given the magnitude of measured surface elevation change over the 3 day period and the correspondence of those measurements to melt measured at ablation stakes, we assume that vertical uplift is a negligible contribution and the point cloud differences represent surface melt. Additionally, each point cloud was independently georeferenced each day, removing complications from horizontal flow.

Bash et al. (2018) measured higher melt rates in active supraglacial streams than on surrounding ice. This higher rate of melt leads to streams downcutting into the glacier surface over time, as long as melt water routing paths remain constant. Bash et al. (2018) also noted the difference in melt rate between an active stream and the nearby channel which the stream formerly occupied. During preliminary analysis of distributed model results, we observed the occurrence of high model errors in areas with visual evidence of surface water flow including thin sheet flow and streams. Further investigation revealed correlation between model errors and the density of linear flow features (designated through DSM analysis and representing streams as

well as thin sheet flow). The northern portion of the study area has predominantly north aspect, which exhibits high density in $W_{20m}$, but low model error. This pattern may result from lower melt on the north aspect, leading to less water production and flow than what is suggested by inferred flow paths in $W_{20m}$. The relation between model underestimation and surface water features suggests the water flowing on the glacier surface is contributing a measurable amount to surface melt. This is consistent with observed patterns of downcutting streams and higher melt in active stream channels, but also shows the importance of water in areas with thin sheet flow over the glacier surface.

Stevens et al. (2018) note the role of kinetic energy in development of surface weathering crusts and the complexity of near-surface glacier hydrology which is only recently coming to light. Temperatures measured in a supraglacial stream on Fountain Glacier in 2014 ranged from 0-0.1°C (St. Germain and Moorman, 2016). These temperatures alone are not sufficient to explain the magnitude of model errors in the vicinity of water features, suggesting that other forms of energy transfer are playing an important role. Idealized crevasse bottom incision equations presented by Fountain and Walder (1998) are driven by discharge rates, suggesting a role in flow related incision for both heat advection as well as energy transfer from kinetic and potential energy. Irvine-Fynn et al. (2011) note that the relative importance of frictional heat contributing to incision rates in non-temperate glaciers remains unresolved. The results of this study suggest that further investigation into energy transfer (heat, kinetic, potential, frictional) and albedo effects from flowing water on non-temperate glaciers is necessary.

The coincidence of high model error with water feature density was noted in areas with thin sheet flow, in addition to the locations of supraglacial streams. The contribution of melt energy to the glacier surface from less concentrated water flow found here has not been examined in previous studies, and these results add new insight to the emerging picture of complex surface hydrology on non-temperate glaciers, as suggested by Stevens et al. (2018). The results of this study suggest that on Fountain Glacier the relative importance of energy flux from water flow is high enough to cause significant errors in the distributed model.

We believe that the relative importance of factors other than solar radiation which are driving melt on Fountain Glacier will have cumulative effects if $ETI_{dist}$ or $EB_{dist}$ were employed for long term modelling. Although this study looks at a short time frame of only 3 days, weather conditions during the study are similar to those found throughout the ablation season on Fountain Glacier and are likely to be representative of average conditions throughout the season. The high melt rates measured in areas of surface water flow, which are absent in both distributed models, are enough to change the terrain characteristics of the glacier over time (as seen in the formation of deeply incised canyons on Fountain Glacier). The changes in terrain feed back into melt dynamics related to slope and aspect, the inability of both models to capture the highest melt rates will slow these feedback mechanisms compared to observed evolution of the glacier surface (St Germain and Moorman, 2019). St Germain and Moorman (2019) show that migration of small surface streams on Fountain Glacier on annual timescales slows or prevents development of deeply incised streams. Similarly, the migration of thin sheet flow over the course of a season is likely to prevent deep heterogeneity across the glacier surface due to this type of water flow. However, the additional melt caused by this type of flow remains unaccounted for in modelling results.

The availability of high resolution melt measurements now allows for analysis of model performance that is not possible with other methods of measuring melt (i.e. ablation stakes or satellite measurements). Building on previous studies of distributed

energy balance models, high resolution distributed melt data may be used to develop simplifications for glaciers similar to Fountain Glacier, which are more appropriate than primarily radiation driven models. Deeply incised canyons such as those found on Fountain Glacier have been noted on 83 other glaciers across the Arctic, including 10 on Bylot Island (St. Germain, 2019). Other studies of glacier terrain characteristics have found that gradual shifts in glacier aspect due to differential melt

have important impacts on total glacier mass balance (Arnold et al., 1996, 2006). This reinforces the importance of developing a model which captures melt driven by both radiation and water flow.

## 5   Conclusions

We present the first study using high resolution measurements derived from SfM and UAV imagery to assess performance of distributed energy balance and enhanced temperature index models, additionally we present a new method of deriving albedo

from the UAV imagery that can be used directly in the grid-based model. The model developed in this study was compared directly to AWS measurements and to distributed measurements on a cell by cell basis. The availability of high resolution data revealed patterns in model performance that could not be found with other traditional methods of measuring glacier melt (i.e. ablation stakes or satellite measurements).

The $ETI_{dist}$ exhibits similar performance at the AWS to other studies implementing ETI models. In the absence of high

resolution distributed data, this model might be applied with confidence across the glacier surface. The bulk performance of the model across the study area is also similar to distributed melt measurements, with a ME similar to the measurement uncertainty. However, the range in melt from both models is an order of magnitude lower than that seen in distributed melt measurements. The lower range in modelled melt combined with spatial patterns found in model error, allowed for investigation of sources of model error.

Significant correlation between surface water flow and model error highlighted the important role of water flow in melt dynamics on Fountain Glacier. Energy transfer from water flow in supraglacial streams is known to cause stream incision and was previously noted by Bash et al. (2018) on Fountain Glacier. The role of thin sheet flow on the glacier surface has not been studied previously, although Stevens et al. (2018) investigate near-surface hydrology and the role of kinetic energy in weather crust development. Our results corroborate the important role of kinetic energy in surface hydrology and suggest

that on Fountain Glacier energy transfer from water flow is a significant driver of melt. Further investigation is necessary to determine the most important contributors of additional energy to the glacier surface from water flow, including the potential role of the lower water albedo when compared to ice.

Given the importance of water flow in the development of deep canyons on Fountain Glacier, a model which accounts for water related energy inputs is necessary to effectively capture long term surface evolution on the glacier. A model which better

reflects the drivers of melt on Fountain Glacier, and potentially similar glaciers across the Arctic, will provide insight into changing melt dynamics in these environments. In future work we hope to extend the time series of melt measurements from UAV imagery and use the distributed dataset to derive a new melt model which captures the melt drivers highlighted in this study.

*Author contributions.* The study was conceptualized by E.A.B., with supervision from B.J.M. Field data collection was designed and performed by E.A.B., with supervision from B.J.M. E.A.B. developed and implemented models, analysed all outputs, and prepared the manuscript. B.J.M. edited the manuscript and provided feedback on analysis.

*Competing interests.* The authors have no competing interests.

5 *Acknowledgements.* This work was supported by a Natural Sciences and Engineering Research Council Discovery Grant. Field support was provided by the Polar Continental Shelf Project, Parks Canada, Northern Scientific Training Program, and the Arctic Institute of North America. E. Bash was supported by the University of Calgary's Eyes High Recruitment Scholarship. We would like to thank Allison Gunther for fieldwork and technical assistance, and Mustafizur Rahman and Shawn Marshall for coding assistance. In addition, we would like to thank Shawn Marshall and two anonymous reviewers for valuable feedback on this manuscript.

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

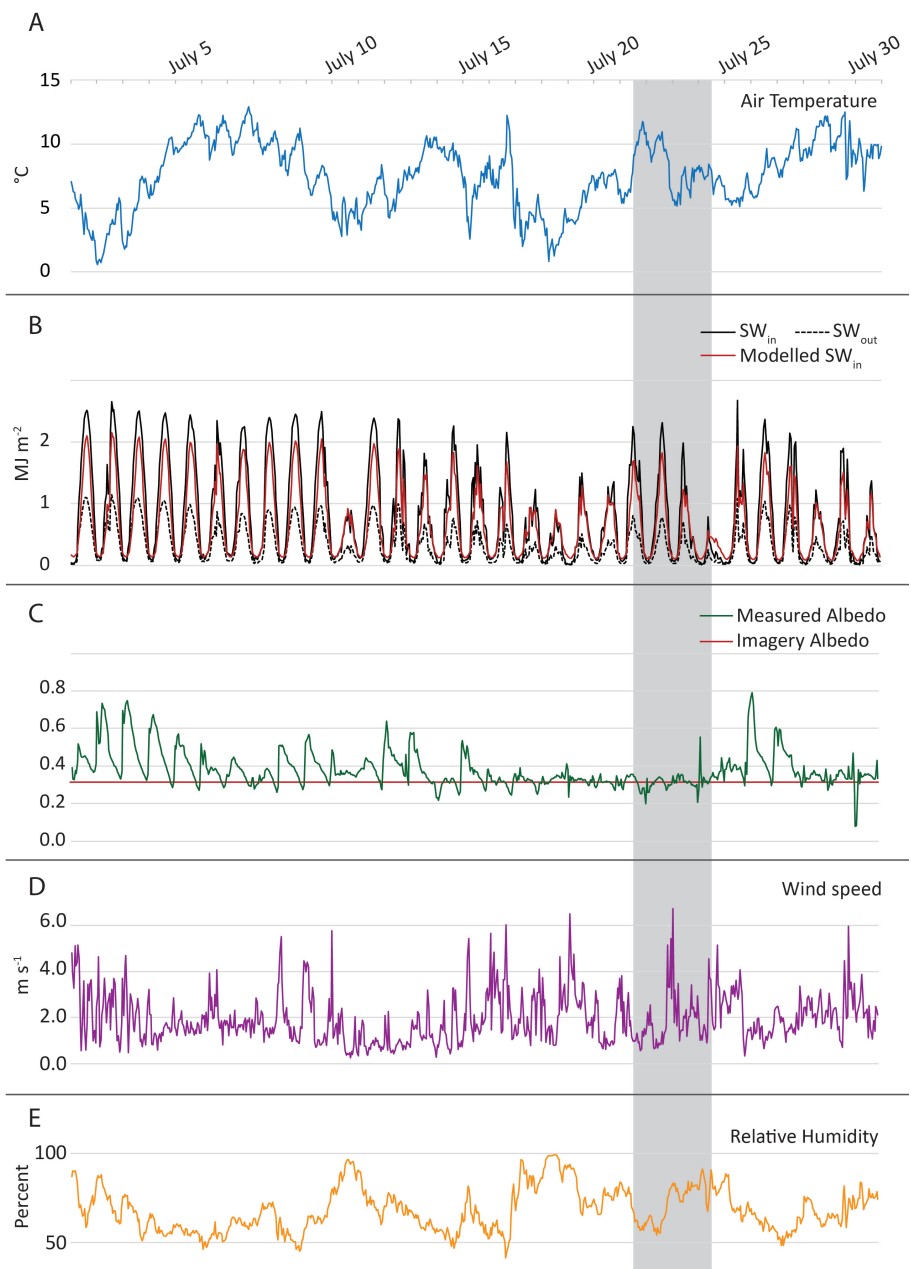

**Figure 2.** Hourly averaged melt model input variables, measured at the AWS for the period July 1-31, 2016. Variables include near surface air temperature (**A**), incoming (SW$_{in}$) and outgoing (SW$_{out}$) shortwave radiation (**B**), albedo (**C**), wind speed (**D**), and relative humidity (**E**). Inputs which are estimated for distributed models are shown in red, incoming shortwave radiation (**A**), and albedo (**B**). The albedo derived from UAV imagery is scaled using the average value at the AWS for July 21-24, so it remains constant throughout the model. The dates of the UAV study are shown in grey.

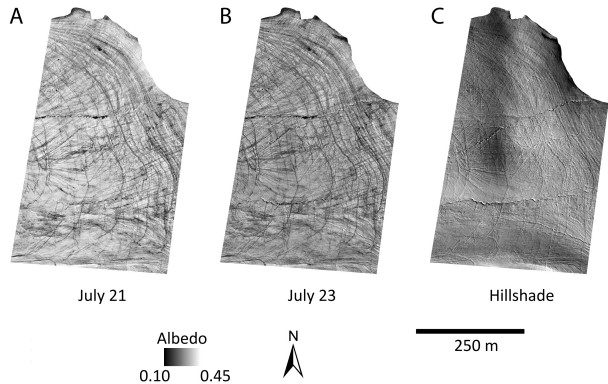

**Figure 3.** Albedo was derived from UAV imagery using an average values from RGB bands and scaling the lower and upper bounds to match the albedo of bare rocks (based on Ryan et al. (2017)) and the average albedo measured on Fountain Glacier during the study period. (**A**) July 21 albedo values. (**B**) Juy 23 albedo values. (**C**) A hillshade model of the study area based on the DSM from July 21.

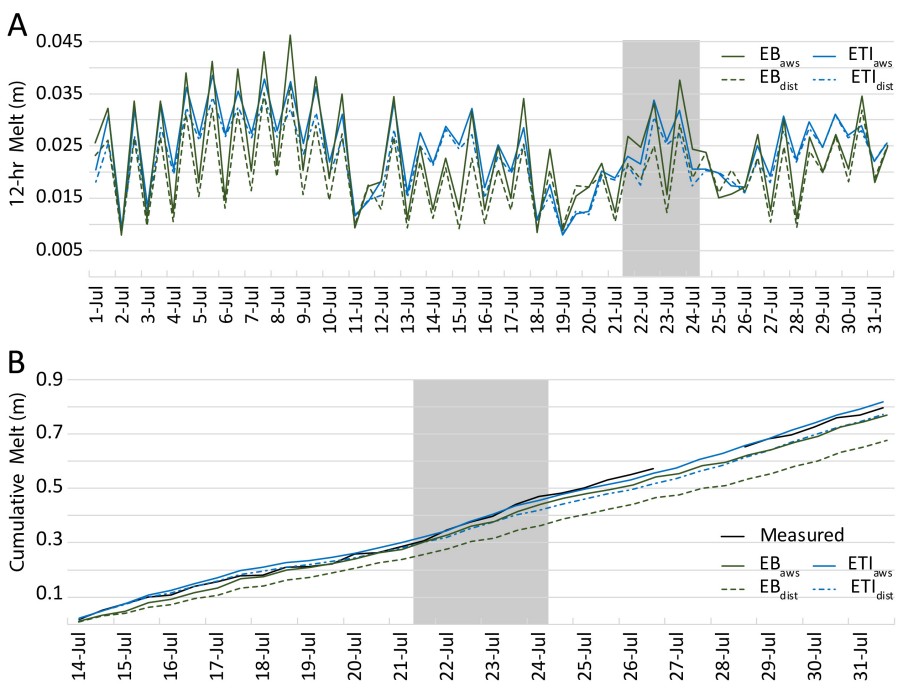

**Figure 4.** (**A**) Modelled 12-hr melt values at the AWS site, July 1-31, 2016. (**B**) Cumulative measured and modelled melt at the AWS, July 14-31, 2016. SR50 measurements were only available for the second half of July, a gap exists in the record due to melt out of the pole holding the SR50. Melt was extrapolated during this time for better visual comparison, extrapolation was based on average melt rates for the day prior to and following the gap. The dates of the UAV study are shown in grey.

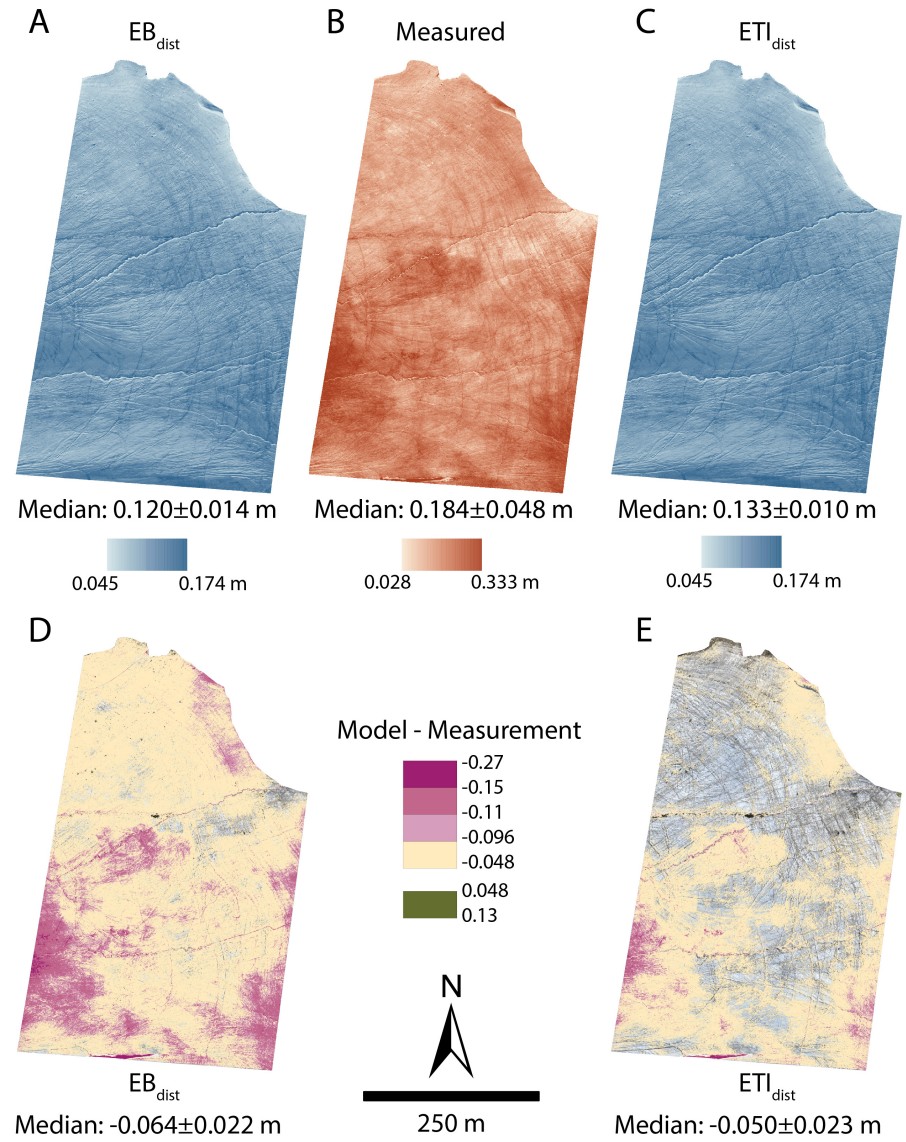

**Figure 5.** Distribution of measured and modelled melt across the study area between July 21 (12:00) and July 24 (12:00). (**A**) Melt estimated using $EB_{dist}$ applied across the study area. (**B**) Melt measured through differencing surfaces reconstructed from UAV imagery. (**C**) Melt estimated using $ETI_{dist}$ applied across the study area. (**D-E**) Model error, calculated by the difference between modelled and measured melt. Cells with an absolute value lower than the measurement uncertainty (0.048 m) are transparent to emphasize areas where the model residuals are high.

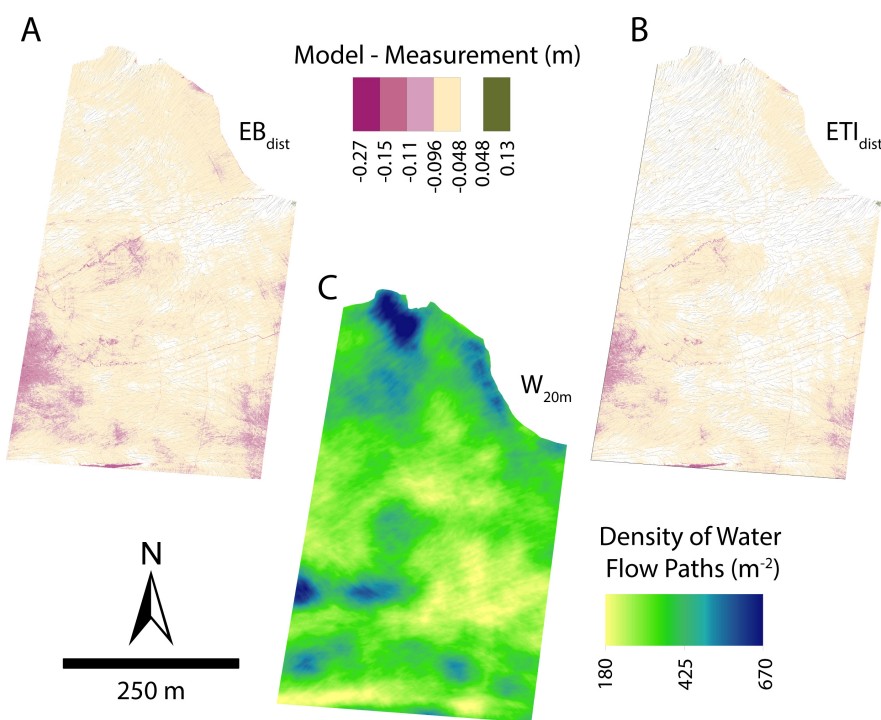

**Figure 6.** Model error with magnitude greater than 0.048 m underlain by flow path features with potential upstream accumulation greater than 15m$^2$ for (**A**) EB$_{dist}$, and (**B**) ETI$_{dist}$. Density of flow accumulation features within a 20 m radius of each cell (**C**).