# Peer review of "Surface melt and the importance of water flow - an analysis based on high-resolution UAV data for an Arctic glacier"

_The Cryosphere, 2019_

## Short Comment (SC1) · 28 May 2019

This is interesting work and the authors get a lot out of a three-day data record. Of course a month or more of data would be really helpful, to increase the signal to noise ratio in both the UAV data and the melt model, but given that this is what the authors have, I am impressed with their inferences. The high-resolution images of surface changes definitely have something to offer to help understand the uncertainty in measurements and models of surface mass balance, especially when it comes to spatial variability at a given elevation and errors in spatial interpolation.

That said, I have some questions about the melt modelling that would be worth considering:

1. The authors have an automatic weather station (AWS) recording what they need to carry out a proper surface energy balance, which could well apply to their study region (with modifications of the absorbed shortwave radiation for slope, aspect, and albedo). Why are the AWS data not invoked, at least to check how much much net energy was available at the AWS site, the associated melt, and whether this is consistent with the UAV data. It is hard to justify a simplified ablation model when the authors have the data to do a complete energy balance, with proper physics. Longwave radiation sounds like its missing, but there are methods to parameterize this as a function of temperature and humidity. This addition would greatly strengthen this manuscript and has the potential to change the results significantly. I understand the basis for a simplified melt model to apply to large, distributed areas where meteorological data are lacking, so the authors could still test their ablation model, but it would be stronger if compared against the 'gold standard' of proper melt physics. The authors should consider a full surface energy balance as a kind of 'ground truthing'.

2. Within the simplified radiation-temperature index melt model that the authors use, I also think that they made an important conceptual error that needs to be corrected. The authors assume that temperature, T, is uniform over the study site while residual temperature, Tr, needs to be adjusted. It is actually the opposite. The whole idea of constructing a decorrelated temperature time series is that net shortwave radiation influences air temperature (a positive correlation). If there is no relation, then Tr = T. Because there is a relation, the solar radiation effect is removed and the 'residual' can be thought of as the 'proxy' for the energy associated with other sources of heat, such as incoming longwave radiation and sensible heat flux. So it is inconsistent to assume that T and absorbed shortwave radiation are correlated at the AWS site, and then to assume that they are not correlated elsewhere (i.e. variable net shortwave radiation with constant temperature). Because there is a relation, the authors' south-facing grid cells, where shortwave radiation is greater, should be associated with warmer temperatures. By assuming a constant temperature everywhere, the authors ends up with lower Tr where net shortwave radiation is high – hence, an underestimate of the melt (consistent with the results). This should be rerun assuming uniform Tr rather than uniform T. This will alter the results and conclusions.

3. I do expect that the interesting correlation with water concentrations on the glacier surface will hold, but the aspect results should change. That said, does the influence of meltwater need to be through kinetic energy? Certainly it is in supraglacial and englacial channels, liberating potential energy for kinetic and thermal dissipation. But another influence for a water film is simply through its albedo effect, water being close to 0.

4. Which brings me to albedo: visually, there appears to be a really large shift in albedo over 1 or 2 days as inferred from the UAV data. This would be hard to explain, the appearance of a systematic darkening of 0.1 or 0.2 over glacier ice, as I read the figure. This is perhaps just the image or illumination angle – reporting actual values here would help. What is the average albedo (and variance) over the study region in these two images, or for each day of the study?

---

## Referee Comment (RC1) · Anonymous Referee #1 · 6 Jun 2019

[Review of manuscript 'TC-2019-81']

[Main Comments]

The authors present a case study of a distributed enhanced-temperature index (ETI) model compared to very high-resolution digital surface models (DSM) for a glacier terminus in Arctic Canada. The authors note a good model performance comparing median changes, though indicate that the model does not capture the full variability of the measurements which are partly associated with surface channels, 'water features' and aspect of asymmetrical banks of surface channels. The study highlights that models are not representing such features, and as such, they must be addressed for future

modelling attempts on Arctic glaciers.

The manuscript is well presented, written and draws upon a great dataset with which to highlight problems of the broad application of ETI models. The work is based upon sound scientific knowledge and the expansion of previously published work using this dataset. While the authors justify their usage of their ETI model, I believe that testing model limitations against a very high-resolution UAV survey requires leveraging available (and local) AWS data for running a full energy balance first, to better understand the processes and errors of the model vs. the measurements. With a greater understanding of the problems facing the modelling efforts on this glacier (and potentially other Arctic glaciers), the authors should then approach the ETI model and demonstrate the significance of their results to the wider science which relies on these simpler approaches.

In places the manuscript also suffers from a lack of justification and reasoning for its methodologies, and in other places, I believe that some key pieces of information are missing. I feel that more analysis, and potentially additional figures may help to strengthen the work. I would very much like to see this work published and I think it is of a high quality for publication in The Cryosphere. I would nevertheless like to see and review the implemented changes. Therefore, the changes I have suggested here and detailed below constitute a major revision.

[Specific comments]

(Page)1-(Line)1: While I think that all the necessary information is here, the first 5-6 sentences should be reworked to improve the flow from short, individual sentences. It currently reads like bullet points.

2-8: Rework the sentence to reflect citations relevant to TI, ETI and EB, separately.

2-12: Check sentence continuity.

2-25: Typo

2-35: NU? I assume Nunavut. Perhaps write out in full for the few instances for the benefit of the reader.

3-10: Include the size of your study area: 0.185km2

3-12: AWS melt I assume refers to a sonic depth gauge? Specify what you mean here... It is measured quantities that you refer to??

3-17: Try to keep consistent with units... Here you move from metres when talking about error to cm for melt rates. These values are also surely in a water equivalent? This was a concern of mine when reading the manuscript as I was unsure if you are comparing UAV differencing (Z difference in m./cm.) and modelled melt (m w.e. /cm w.e.).

3-25: You describe your methodology for pole tilt of the SDG, but the dates of your ETI model (at both point and distributed) extend beyond the UAV measured differences with no real justification for why... perhaps I've missed something here or it hasn't been explained.

3-30: "30" what??

3-31: I'd like to see some reasoning for the selection of your chosen methodology for dealing with SDG data averages and gaps. Why the specified interval? Does it affect your results?

4-4: Whilst this work is published in Bash et al. (2018) and doesn't all need to be re-peated, I would like to see a few of the core details and justifications for methodologies of the SfM model construction... For example, why M3C2, and not 2.5d differencing was considered. A short reminder here would be useful.

4-7: Is interpolation justified here? With what method? Why were their gaps in your DSM?

6-7: So Iabs is SWnet? Perhaps change this terminology to be clear and consistent

with the literature.

6-8: What are your derived TF and SRF values?! This is important to mention somewhere. Are your values believable? How do they compare to the other TI, ETI models (as you compare to in your discussion)?

6-9: I need some justification for modelling over these dates, and not just the DSM differencing period (3 days).

6-11: Have some tests of the 0C threshold been performed? This threshold can be rather variable. I believe that optimising this threshold could be beneficial. This holds then to my major comment regarding the model. I believe the data and locality of the AWS makes it highly desirable to perform an energy balance approach for the point-scale, if not for both point and distributed models, to aid process understanding and support the application of your ETI approach. The authors have a lot of valuable data with which to calibrate/validate their model at the distributed scale (such as stakes and surface information from the UAV) with which to provide the best possible ETI model. Propagating the model error then with the DSM error can provide a strengthened analysis of how, in the best case scenario, the ETI succeeds/fails to capture important information about glacier mass loss.

6-12: To distribute model variables. . . reword here.

6-13: Although I agree that differences will likely be small, I think it valuable to distribute your temperatures in your model domain, as it may still vary enough to influence your 0C assumptions for melt onset. This also draws in to the limits of the residual approach you adopt, as mentioned in the open discussions by Professor Marshall. I think a simple environmental lapse rate would suffice for the small elevation range you describe.

6-15: This is modelled using the surface topography taken from the first DSM right? Specify that here.

6-16: Another interesting and important aspect to test would be the effect of cell resolution. A 0.1 m resolution is incredible and interesting to test what models can and cannot do, but is not realistic for any level of glacier-scale or regional modelling. I think it relevant to have some level of testing, and discussion regarding this. For example, the ETI fails to capture the variability seen in the DSMs, but is this consistent at 1 m, 10 m, 30 m?

6-18: Provide minimum solar angles.

6-26: Check sentence continuity.

6-27: Your figure 2 implies that you have a net radiometer (or rather both up-facing and down-facing pyranometers) to derive your albedo at the point-scale. How does this compare to your albedo map derived from your RGB histogram stretch from 0.55 upward? What are the ranges of albedo derived? Also, the values seem largely different between 3 days when looking at figure 3. It looks as though the whole domain darkened by 0.05-0.1. Perhaps this was some effect of the cloud filtering you performed (according to Bash et al., 2018)?

7-1: Still need some reasoning for your model period selection.

7-10: Measured melt from the DSM? Again is this a Z-difference or ablation? Are you truly comparing the two here? I'm sure your model does not calculate vertical lowering, but melt (in w.e. units). Are you converting your DSM differences with density values of ice?? What values? Measured or assumed?

7-11: Again, please provide some justification for NMAD and ME over other metrics.

7-18: The model was run to quantify surface water production? Or just a 'watershed' analysis? Perhaps add a map of this to Figure 3?

8-9: The lower variability of the modelled values are not surprising, given that the model has two variables to consider for a system that is far more complex.

10-2: Figure 5A4-D4 does not show the correlation, but area of water features. I also

fail to see how the errors of the AWS and distributed models are linked here. Perhaps include a correlation figure somewhere.

10-21: What uncertainties? Model uncertainties are not considered against your model-DSM comparisons. Your model error is simply the difference with the observed values.

10-22: Check citation format.

10-27: what do you mean by muted?

10-32: Again, interested to see the effect of model resolution on your results as it would be very relevant for future work.

11-5: And Horizontal motion?

11-12: Again to see another figure with some correlations would be nice.

11-17: Could this be a result of measurement uncertainty? Lighting from the processing of the SfM images? The processing of clouds and histogram stretching applied in your former paper? Although predominantly south facing, roughness may play a role here, which is of course not considered by ETI models.

11-21: Have you considered any uncertainty due to SfM model construction for steep sided relic stream features that you mention? Did you obtain any oblique images? Is there anything worth noting about that here?

12-16: Interesting that Figure 2 doesn't show such a dampened cycle. Please see suggestions for Figure 2. What about Wind speed? (again plot in Figure 2)... A temperature factor will under-represent the contribution from turbulent fluxes if your glacier terminus is heavily affected by strong katabatic winds. Again, anything worth noting here?

13-3: The conclusion reads a bit like a discussion in places.

13-5: ETI 'model'. The albedo sentence feels a little out of place and should be reformulated into the previous sentence.

13-24: Add citations here.

13-26: More pronounced compared to which studies?

[Figures]

Figure 2: Perhaps use colour here to aid the visual differences between measured and modelled point-based melt (or surface lowering??). For panel C. What do you mean by distributed variables? Surely the values are the same, as you are comparing the melt/surface lowering at the AWS? The variables require no distribution and surely are the same as panel B? Why not compare the measured and modelling SWin on the same panel to give confidence to the reader that your modelled radiation, based upon surface topography, is valid? Also, I would suggest converting units to Wm-2, for consistency with much of the literature, and to compare with other studies. As suggested in the specific comments above, I think it would be valuable to demonstrate the full EB data, including wind speed (and direction if there is anything interesting to say about katabatic winds and its directional consistency) as well as relative humidity and LW if available. Equally, showing the sub-period of the UAV DSM acquisitions is key!

Figure 4: Your model error is Modelled minus Measured? So your negative values are under-estimation? Please clarify your convention. Transparent areas don't show where the model performs poorly, but rather where modelled-measured differences are within the error range of the measurements. What about model error estimates here? These should be propagated.

---

## Referee Comment (RC2) · Anonymous Referee #2 · 7 Aug 2019

Review of "Assessing the performance of a distributed radiation-temperature melt model on an Arctic glacier using UAV data" by Bash & Moorman

General Comments

This study draws upon high-resolution glacier surface melt estimates to validate and test the robustness of an enhanced temperature index melt model across the terminus of a glacier in Arctic Canada. While the study draws heavily upon previous work by the authors, the novelty of this particular study is clear as one of the first opportunities to investigate the performance of a distributed melt model at high spatial resolution, allowing for insights into the topographic controls on model performance. Overall, it has

the potential to be an impactful and citeable study for glacier mass balance and energy balance researchers by providing new insights into the performance of distributed melt models, and the comments here primarily focus on strengthening the arguments of the authors and improving the clarity in places:

1) There are several instances where the authors draw upon the methodologies of other studies to guide their ETI model development. However, what is often lacking is a simple description of the approach used in the cited study, and a statement explaining why this methodology was chosen by the authors. Examples of cited works that would benefit from deeper explanation include: Goswami et al. (2000); Bugler (1977); Rippin et al. (2015); Höhle and Höhle (2009)

2) A brief statement should be made early in the paper indicating whether reported melt values (m) are in ice equivalent or water equivalent (i.e. have you converted your surface elevation change observations to melt equivalent, or vice versa).

3) The determination of albedo is somewhat concerning, or at least the section describing the determination of albedo requires more detail and justification for the reader. It would be my impression that the scaling of surface albedo needs to be done twice, once for each orthoimage, with scaling being conducted using the albedo measured at the AWS at the time of the survey as validation for that cell. An off-ice site (rock or debris) could be used as a location of assumed constant albedo of ∼0.1. Instead, it is unclear why the authors would use the AWS-derived average albedo (rather than maximum) as the upper limit for their albedo threshold across the area of interest on both survey days. Hopefully revisiting the albedo calculations might also help explain the surface darkening noticed by another commenter on the discussions page! Plotting the progression of surface albedo at the AWS over the study period could help confirm whether this darkening is real.

4) A more detailed error analysis, and clearer reporting of errors and uncertainty, would significantly strengthen this work. Currently "model error" is used to describe the residual values between modelled (ETI_dist) and "measured" (see note below) melt at each cell. However, each of the parameters included in the model should have an associated (or at the very least estimated) uncertainty, as will the DSM models and calculated melt from DSM differencing.

- To help with this, and to strengthen your argument for correlation between model errors and specific parameters, it could be worthwhile to include a figure with four plots that show model error vs. 1) aspect (with a full range from 0-360°), 2) slope, 3) albedo, and 4) density of water features, by extracting these cell values over the entire region.

5) On a semantics note, the differencing of DSM models to determine melt is not a melt "measurement" but rather it is a calculation, or estimate (considering that the DSMs have their own model uncertainties). It is fair to treat the DSM melt estimates as a measurement for the sake of ETI model validation, but the authors should briefly acknowledge this and be clear about the source and magnitude uncertainties associated with this "measurement."

6) One notable gap is an acknowledgement of, or discussion about, how parameters like aspect and slope are incorporated, or fail to be incorporated, into ETI_dist through the estimation of distributed radiation using the Goswami et al. (2000) approach. There is a good opportunity here to explore the sensitivity of ETI_dist to slope and aspect!

Specific/minor comments:

P1-L19: . . .14% of the world's glacier ice. . . ← specify if this is area or volume

P2-L13: Available measurements. . . The wording of this sentence is awkward, rewrite or combine with previous sentence to help with clarity.

P2-L25: . . .in high temporal and spatial detail. . . (replace great)

P3-L13-16: Regarding ablation stakes, I am curious why you did not also use these 17 ablation stake observations as validation for ETI_dist, in the same way you use the AWS as a point observation. Perhaps you did not have stake measurements on

the days of the UAV survey... However, I am perplexed by the reporting of an RMSE for the stake observations, perhaps you mean the RMSE of the modelled melt (from DSM differencing) at the stake locations? It is important to be clear that your DSM differencing is not communicated "measured" melt, but rather modeled, or estimated, melt using the geodetic method.

P3-L23: Suggest including the manufacturers name for the SR50.

P3-L31-32: Please provide some justification for why a 5-hour rolling average was chosen, and how samples qualified as being "significantly different" from the population.

P4-L3-4: When referring to resolutions (0.10 m, 0.02 m) specify whether these are horizontal resolution (your cell size) or vertical resolution; also be specific with the reported uncertainty – which is presumably vertical uncertainty? +/-?

P6-L13: "...temperature was assumed to remain constant across the area." Out of curiosity, do you have a rough idea of what the temperature gradients are in this area? A quick correlation analysis between model error and surface elevations from your last DSM would be a simple way to verify that this assumption is reasonable.

P6-L15: What do you mean by "modify"? I would suggest giving much more attention to describing how the slope and aspect correction is applied in ETI_dist, particularly given the correlation of your error to aspect later on (see general comment #7)!

P6-L19-30: This is the section where it would be helpful to have some more detail from the previous studies you draw upon build your model and conduct corrections. This section really describes the heart of your model, and any additional detail you can provide will help readers understand the reasoning for your model design and why it performs the way it does. I might even suggest creating a simple flow-chart that illustrates the model inputs and their sources.

P6-L27: Is there a reason your reported range in albedo goes from high to low? (Rather than "0.1-0.55"?)

P7-L12: As noted in the general comments, a brief explanation of the Höhle and Höhle paper's approach would be helpful – I am personally curious why the median is used instead of the mean!

P7-L19: Why was 1500 cells chosen as the threshold for number of cells contributing to a water flow feature, and can you express this value in the area equivalent (e.g. square meters)?

P8-L8: Uncertainties should be included the modelled and measured values reported here.

P9-L4: Be clear what this correlation value is (Pearson correlation coefficient) or by using r = 0.34. Putting the correlation value at the end of this sentence seems awkward, maybe try rewording?

P9-L7: Similar to above, correct to include "... much lower (r < 0.1, and p > ___)..."

P10-L1: Suggest observation rather than "experience"

P10-L3: "-0.048 m"; also reword for clarity and include r =.

P10-L15: "...deviation of 0.00083 m h-1, which is similar to other..."

P10-L20: Be specific, which year?

P10-L32: Specify, horizontal or vertical resolution

P11-L4: "offsets"

P11-L12: "Correlation between aspect..." This sentence feels out of place in a description of work by Bash et al. (2018), perhaps try rewording or open this paragraph with this sentence or something similar. E.g. "The correlation (r = X) between aspect and model error ..."

P11-L18: Tighten up wording, e.g. "Bash et al. (2018) measured higher melt rates in active supraglacial streams than on the surrounding ice."

[Figure]

P11-L23: "... between the density of linear..."

P12-L9-16: It is unclear what this paragraph contributes to the paper here, rather it seems to interrupt a discussion of water flow features and their impact on melt. Perhaps add more context, otherwise remove.

P12-L20: "...have a greater relative importance."

P12-L31: What do you mean by "simplifications"?

P13-L5: Can you be specific about how you actually build upon the work of Rippin et al. (2015) to estimate albedo? Is this by introducing a scaling approach?

P13-L10: Unclear what this first sentence is saying regarding "other implementations"

P13-L19: It is not clear how Stevens et al. (2018) and the development of weather crusts relates to this study. Either take more time to explain the relevance (in the discussion section) or remove from conclusions.

Figure comments

Figure 2. Extending the E and F y-axis down to zero would help your arguments in the text. Check consistency with bold-type for your graph subsets and parenthesis e.g. A) vs (B), and correct formatting of Iin in the 2nd last line. Can you also explain the gap at the beginning of (F)?

Figure 4. Recommend including the study dates – "... across the study area between July 21 (hh:mm) and July 23 (hh:mm)."

---

## Author Comment (AC1) · 3 Sep 2019

Thank you for your comments, you bring up several good points. Our response is below:

1. While we had considered comparing the UAV measurements with an energy balance model, our main goal in the study was geared towards a more broadly applicable model. However, your point of using the energy balance as another kind of ground truth is valid and we will endeavor to include this in the revised manuscript.

Regarding the comparison of UAV data to AWS data, this was presented in a previous

manuscript, but we acknowledge that reiterating it here would also be valuable.

2. This is an insightful comment on the physical underpinning of our statistical model, which we had not considered. We will adjust the temperature index model to hold Tr constant and allow T to vary, and report updated results in the revised manuscript.

3. The manuscript as posted in the discussion forum has already been updated to discuss other potential mechanisms for water flow leading to increased melt. Although this may include an influence from albedo, the weak relationship between albedo and model error suggests that is not the main driver for increased melt in these locations. We have acknowledged that the actual mechanism has not been studied and thus we can only speculate on the possible causes for the increased melt where surface water is present.

4. Indeed there is an overall decrease in the gridded albedo between July 21 and July 23, which is likely due to differences in imagery between the two days, in addition to real lowering of albedo which was recorded at the AWS. The average difference between the two days across the study area is 0.07, but we believe this translates to a minor potential error in the melt calculations. We will address the implications of the albedo more thoroughly in the revised manuscript.

---

## Author Comment (AC2) · 3 Sep 2019

Thank you for your comments and edits of our manuscript. We agree that the inclusion of an energy balance model will strengthen the analysis. Following your suggestion and that of Dr. Marshall, we will develop a local energy balance model and include this in the updated manuscript.

In the revised version of the paper we will carefully consider all further points you raised and try to incorporate them into the document.

---

## Author Comment (AC3) · 3 Sep 2019

Thank you for your comments on this manuscript. Please find below our responses to your general comments. In the updated manuscript, we will consider all further points you raised and try to incorporate them.

1. We agree that in several places the manuscript will benefit from better explanation of our methods and choices. We will specifically address the references you point out, but also look for other opportunities to further clarify the modelling methods.

2. We will state this explicitly.

[Figure]

3. The scaling was indeed performed once for each image, and we will clarify this in the text. Regarding the choice of scaling parameters, the darkest pixels in the image do correspond to surface debris on the glacier, and thus the lower end of the albedo spectrum is fixed to those locations. We chose this approach in favor of a particular off-ice location because we have not measured albedo off ice and can't say with certainty that the albedo of any particular location is 0.1. With respect to the upper end of the spectrum, we selected the average AWS albedo for several reasons: 1) because the AWS location is relatively clean ice it is likely representative of the highest albedo in the study area; 2) because the imagery was taken at a particular point in time it reflects an instantaneous (almost) albedo, but must be applied over multiple days, we feel an average is a better representation of daily surface characteristics; 3) because albedo varies with solar angle we were hesitant to assign the maximum measured albedo to the highest values as it would likely over estimate albedo.

4. We will include a further discussion of the estimated uncertainty associated with model parameters. We originally considered the inclusion of scatter plots as you suggest, but given that the dataset include millions of points, these figures tend to be too noisy to visualize patterns effectively. It is a problem with UAV data, which is difficult to address.

5. We will elaborate on the sources of uncertainty in the calculation of surface lowering from UAV data and clarify further the use of this calculation as a ground truth, which we refer to as a measurement.

6. We agree that the radiation methods could be strengthened by further discussion. We have performed some preliminary sensitivity analysis in exploring different modelling options for solar radiation and will include some discussion of this in the revised manuscript.
* * *

---

## Author Comment (AC4) · 5 Sep 2019

We would like to add an additional response, regarding the time frame for the modelling (comments 3-25, 6-9, 7-1) and effects of grid resolution. With the availability of AWS data between July 13 and August 2 we chose to split the data into a training period and validation period. Data from the period July 13-July 21 as training data for the ETI model, while data from July 21-August 2 is used to as validation for the model. Even though the main focus of the work is on the comparison with distributed data from Bash et al. (2018), the opportunity exists for a longer comparison period at the AWS. We felt that this strengthened the examination of the model, rather than simply looking at the

short time frame where distributed measurements are available.

The ETI formulation based on distributed model inputs was used over the entire grid, which includes the AWS location. Although the distribution of temperature and radiation is based on measurements at the AWS, modelled values for radiation and residual temperature differ due to slight differences in the DSM cell containing the AWS and the true orientation of the instruments (i.e. the radiometer is levelled, but the grid cell is not completely level). For this reason we also included the modelled radiation in Figure 2.

We will make this decision clearer in the text of the manuscript.

Finally, regarding the suggestion of investigating the effects of resolution on the model results - we agree that this would be interesting and potentially useful, but feel it is outside the scope of the present study, which is focused on the comparison to melt derived from UAV imagery. The effects of resolution have been investigated in detail by:

Irvine‐Fynn, T. D., Hanna, E., Barrand, N. E., Porter, P. R., Kohler, J., & Hodson, A. J. (2014). Examination of a physically based, high‐resolution, distributed Arctic temperature‐index melt model, on Midtre Lovénbreen, Svalbard. Hydrological Processes, 28(1), 134-149.

Hopkinson, Chris, et al. "The influence of DEM resolution on simulated solar radiation‐induced glacier melt." Hydrological Processes: An International Journal 24.6 (2010): 775-788.

Arnold, Neil, and Gareth Rees. "Effects of digital elevation model spatial resolution on distributed calculations of solar radiation loading on a High Arctic glacier." Journal of Glaciology 55.194 (2009): 973-984.

We can include a discussion of the potential influences of resolution on our results in the context of previous work on the subject of resolution.

---

## Author Response (AR1)

Dear Dr. Farinotti and Reviewers,

Thank you for your feedback on this manuscript. We have attempted to address all the points brought up in the reviews, this has resulted in several major changes to the modelling contained in the paper and thus the results and presentation. The following major changes were made:

1. As per the suggestion of Shawn Marshall and reviewer #1 we have changed the way the ETI model is distributed across the study area. Specifically, the residual temperature (representing melt processes other than shortwave radiation) is now kept constant across the grid. This is effectively the same as distributing the temperature based on equations 10 & 11 (described on P9). This has not substantially changed the results, but did resolve issues with poorly modelled melt on south aspects.
2. We have added an energy balance model and run four model formulations (both point and distributed models) for the month of July 2016 to compare the performance of the models against one another, and also against AWS data.
3. As per the suggestion of reviewer #2, we have revisited the methods for deriving albedo from orthoimages. Values are now scaled by fixing the value of the AWS cell to be the average measured between July 21-24, rather than assigning the average of July 13-31 as the upper end. This means that albedo in the AWS grid cell is constant throughout the model runs. We have tried to better explain our methods in text, and have also addressed the slight darkening between July 21 and 23.
4. During our revision of the models we discovered an indexing problem in the solar radiation model. Correcting this problem has dramatically improved the radiation values in the model.
5. The vast number of data points in UAV datasets make teasing out relationships difficult. We are constantly trying to improve the way we address this and have adopted a new strategy using robust statistical measures to calculate correlations. We acknowledge that both reviewers suggested adding figures showing the relationships between model error and aspect, slope, and water flow. The number of data points we are dealing with (>18 million) makes these type of figures unintelligible, so we have chosen to leave them out. Instead we have added a table of correlation statistics and a discussion of the difficulty of using traditional statistical measures in this study.

Below are our responses to the specific comments of the reviewers. Page and line numbers in our response refer to the track changes version of the document.

Thank you,
Eleanor Bash

Reviewer 1 comments:

(Page)1-(Line)1: While I think that all the necessary information is here, the first 5-6 sentences should be reworked to improve the flow from short, individual sentences. It currently reads like bullet points.

Done. P2-L1-7

2-8: Rework the sentence to reflect citations relevant to TI, ETI and EB, separately.

Done. P2-L16-17

2-12: Check sentence continuity.

Done. P2-L20

2-25: Typo

Done. P2-L35

2-35: NU? I assume Nunavut. Perhaps write out in full for the few instances for the benefit of the reader.

Done throughout.

3-10: Include the size of your study area: 0.185km2

Done. P3-L22

3-12: AWS melt I assume refers to a sonic depth gauge? Specify what you mean here. . . It is measured quantities that you refer to??

Clarified. P3-L25

3-17: Try to keep consistent with units. . . Here you move from metres when talking about error to cm for melt rates. These values are also surely in a water equivalent? This was a concern of mine when reading the manuscript as I was unsure if you are comparing UAV differencing (Z difference in m./cm.) and modelled melt (m w.e. /cm w.e.).

Units have been changed to be consistent throughout.

3-25: You describe your methodology for pole tilt of the SDG, but the dates of your ETI model (at both point and distributed) extend beyond the UAV measured differences with no real justification for why. . . perhaps I've missed something here or it hasn't been explained.

Clarified. P4-L8

3-30: "30" what??

Clarified. P5-L4

3-31: I'd like to see some reasoning for the selection of your chosen methodology for dealing with SDG data averages and gaps. Why the specified interval? Does it affect your results?

More description of methods has been added. P5-L4-11

4-4: Whilst this work is published in Bash et al. (2018) and doesn't all need to be repeated, I would like to see a few of the core details and justifications for methodologies of the SfM model construction. . . For example, why M3C2, and not 2.5d differencing was considered. A short reminder here would be useful.

Detail added. P5-L18-19

4-7: Is interpolation justified here? With what method? Why were their gaps in your DSM?

Detail added. P5-L20-21

6-7: So Iabs is SWnet? Perhaps change this terminology to be clear and consistent with the literature.

Changed throughout.

6-8: What are your derived TF and SRF values?! This is important to mention somewhere. Are your values believable? How do they compare to the other TI, ETI models (as you compare to in your discussion)?

A table of values has been added as well as a brief discussion of their relationship to other studies. P8-L27-P9-L2

6-9: I need some justification for modelling over these dates, and not just the DSM differencing period (3 days).

Clarified. P9-L4-6

6-11: Have some tests of the 0C threshold been performed? This threshold can be rather variable. I believe that optimising this threshold could be beneficial. This holds then to my major comment regarding the model. I believe the data and locality of the AWS makes it highly desirable to perform an energy balance approach for the pointscale, if not for both point and distributed models, to aid process understanding and support the application of your ETI approach. The authors have a lot of valuable data with which to calibrate/validate their model at the distributed scale (such as stakes and surface information from the UAV) with which to provide the best possible ETI model. Propagating the model error then with the DSM error can provide a strengthened analysis of how, in the best case scenario, the ETI succeeds/fails to capture important information about glacier mass loss.

Consideration of the implications of the threshold has been added. P8-L24-28

6-12: To distribute model variables. . . reword here.

This has changed substantially due to new model descriptions.

6-13: Although I agree that differences will likely be small, I think it valuable to distribute your temperatures in your model domain, as it may still vary enough to influence your 0C assumptions for melt onset. This also draws in to the limits of the residual approach you adopt, as mentioned in the open discussions by Professor Marshall. I think a simple environmental lapse rate would suffice for the small elevation range you describe.

See major revision comment 1.

6-15: This is modelled using the surface topography taken from the first DSM right? Specify that here.

Clarified. P7-L10

6-16: Another interesting and important aspect to test would be the effect of cell resolution. A 0.1 m resolution is incredible and interesting to test what models can and cannot do, but is not realistic for any level of glacier-scale or regional modelling. I think it relevant to have some level of testing, and discussion regarding this. For example, the ETI fails to capture the variability seen in the DSMs, but is this consistent at 1 m, 10 m, 30 m?

As per our previous response, we agree this would be a valuable analysis, but feel it would detract from the overall goal of the paper.

6-18: Provide minimum solar angles.

Done. P7-L15

6-26: Check sentence continuity.

Rewritten. P7-L26-28

6-27: Your figure 2 implies that you have a net radiometer (or rather both up-facing and down-facing pyranometers) to derive your albedo at the point-scale. How does this compare to your albedo map derived from your RGB histogram stretch from 0.55 upward? What are the ranges

of albedo derived? Also, the values seem largely different between 3 days when looking at figure 3. It looks as though the whole domain darkened by 0.05-0.1. Perhaps this was some effect of the cloud filtering you performed (according to Bash et al., 2018)?

A more detailed discussion of the albedo has been added. P8-L6-11

7-1: Still need some reasoning for your model period selection.

Clarified. P9-L14-16

7-10: Measured melt from the DSM? Again is this a Z-difference or ablation? Are you truly comparing the two here? I'm sure your model does not calculate vertical lowering, but melt (in w.e. units). Are you converting your DSM differences with density values of ice?? What values? Measured or assumed?

For consistency with SR50 measurements and M3C2 differences, the ETI model output is in m of surface lowering (since it is based on a statistical relationship with the SR50 measurement). The EB output is in m ice equivalent (not water). We have made the units more explicit in text.

7-11: Again, please provide some justification for NMAD and ME over other metrics.

A deeper discussion of statistics has been included. P9-Section 2.4

7-18: The model was run to quantify surface water production? Or just a 'watershed' analysis? Perhaps add a map of this to Figure 3?

Clarified. A map of this density and the linear potential accumulation features has been added in a new figure (Figure 6) P10-L15-21

8-9: The lower variability of the modelled values are not surprising, given that the model has two variables to consider for a system that is far more complex.

Agreed, however, the EB model shows the same patterns.

10-2: Figure 5A4-D4 does not show the correlation, but area of water features. I also fail to see how the errors of the AWS and distributed models are linked here. Perhaps include a correlation figure somewhere.

We have updated the final figure, which we hope makes it more useful to see the relationship between high error and W20m.

10-21: What uncertainties? Model uncertainties are not considered against your model-DSM comparisons. Your model error is simply the difference with the observed values.

We have expanded our discussion of uncertainty on P12-L2-10

10-22: Check citation format.

This sentence was removed.

10-27: what do you mean by muted?

Rewritten. P14-L7

10-32: Again, interested to see the effect of model resolution on your results as it would be very relevant for future work.

11-5: And Horizontal motion?

Added. P14-L20

11-12: Again to see another figure with some correlations would be nice.

11-17: Could this be a result of measurement uncertainty? Lighting from the processing of the SfM images? The processing of clouds and histogram stretching applied in your former paper? Although predominantly south facing, roughness may play a role here, which is of course not considered by ETI models.

After adjusting the model, this relationship has become negligible.

11-21: Have you considered any uncertainty due to SfM model construction for steep sided relic stream features that you mention? Did you obtain any oblique images? Is there anything worth noting about that here?

The streams within the study area are less than 1m deep. The larger deeply incised canyons are outside the study area.

12-16: Interesting that Figure 2 doesn't show such a dampened cycle. Please see suggestions for Figure 2. What about Wind speed? (again plot in Figure 2). . . A temperature factor will under-represent the contribution from turbulent fluxes if your glacier terminus is heavily affected by strong katabatic winds. Again, anything worth noting here?

This paragraph has been removed as suggested by reviewer 2.

13-3: The conclusion reads a bit like a discussion in places.

We have updated the conclusion with this in mind.

13-5: ETI 'model'. The albedo sentence feels a little out of place and should be reformulated into the previous sentence.

Done. P16-24

13-24: Add citations here.

This sentence was removed.

13-26: More pronounced compared to which studies?

See above.

[Figures]

Figure 2: Perhaps use colour here to aid the visual differences between measured and modelled point-based melt (or surface lowering??). For panel C. What do you mean by distributed variables? Surely the values are the same, as you are comparing the melt/surface lowering at the AWS? The variables require no distribution and surely are the same as panel B? Why not compare the measured and modelling SWin on the same panel to give confidence to the reader that your modelled radiation, based upon surface topography, is valid? Also, I would suggest converting units to Wm-2, for consistency with much of the literature, and to compare with other studies. As suggested in the specific comments above, I think it would be valuable to demonstrate the full EB data, including wind speed (and direction if there is anything interesting to say about katabatic winds and its directional consistency) as well as relative humidity and LW if available. Equally, showing the sub-period of the UAV DSM acquisitions is key!

Figure 2 has been updated to include color, and the other inputs of the energy balance model.

Figure 4: Your model error is Modelled minus Measured? So your negative values are under-estimation? Please clarify your convention. Transparent areas don't show where the model performs poorly, but rather where modelled-measured differences are within the error range of the measurements. What about model error estimates here? These should be propagated.

We have clarified our convention in text

Reviewer 2 comments:

1) There are several instances where the authors draw upon the methodologies of other studies to guide their ETI model development. However, what is often lacking is a simple description of the approach used in the cited study, and a statement explaining why this methodology was chosen by the authors. Examples of cited works that would benefit from deeper explanation include: Goswami et al. (2000); Bugler (1977); Rippin et al. (2015); Höhle and Höhle (2009)
We have added further detail to the methods used from these references P7, P10

2) A brief statement should be made early in the paper indicating whether reported melt values (m) are in ice equivalent or water equivalent (i.e. have you converted your surface elevation change observations to melt equivalent, or vice versa).
We have indicated the measurement units on P6-4, P8-L15

3) The determination of albedo is somewhat concerning, or at least the section describing the determination of albedo requires more detail and justification for the reader. It would be my impression that the scaling of surface albedo needs to be done twice, once for each orthoimage, with scaling being conducted using the albedo measured at the AWS at the time of the survey as validation for that cell. An off-ice site (rock or debris) could be used as a location of assumed constant albedo of ~0.1. Instead, it is unclear why the authors would use the AWS-derived average albedo (rather than maximum) as the upper limit for their albedo threshold across the area of interest on both survey days. Hopefully revisiting the albedo calculations might also help explain the surface darkening noticed by another commenter on the discussions page! Plotting the progression of surface albedo at the AWS over the study period could help confirm whether this darkening is real.
See comment above addressing changes to the albedo parameterization

4) A more detailed error analysis, and clearer reporting of errors and uncertainty, would significantly strengthen this work. Currently "model error" is used to describe the residual values between modelled (ETI_dist) and "measured" (see note below) melt at each cell. However, each of the parameters included in the model should have an associated (or at the very least estimated) uncertainty, as will the DSM models and calculated melt from DSM differencing.
We have expanded our discussion of the uncertainty. P10-L2

- To help with this, and to strengthen your argument for correlation between model errors and specific parameters, it could be worthwhile to include a figure with four plots that show model error vs. 1) aspect (with a full range from 0-360∘ ), 2) slope, 3) albedo, and 4) density of water features, by extracting these cell values over the entire region.

5) On a semantics note, the differencing of DSM models to determine melt is not a melt "measurement" but rather it is a calculation, or estimate (considering that the DSMs have their own model uncertainties). It is fair to treat the DSM melt estimates as a measurement for the sake of ETI model validation, but the authors should briefly acknowledge this and be clear about the source and magnitude uncertainties associated with this "measurement."
We have included more detail about the work of Bash et al. 2018. And explicitly acknowledged the assumption that melt and surface lowering are equivalent. P3-L29

6) One notable gap is an acknowledgement of, or discussion about, how parameters like aspect and slope are incorporated, or fail to be incorporated, into ETI_dist through the estimation of distributed radiation using the Goswami et al. (2000) approach. There is a good opportunity here to explore the sensitivity of ETI_dist to slope and aspect!

Specific/minor comments:

P1-L19: . . .14% of the world's glacier ice. . . ← specify if this is area or volume
Done. P2-L1
P2-L13: Available measurements. . . The wording of this sentence is awkward, rewrite or combine with previous sentence to help with clarity.
Rewritten. P2-L20
P2-L25: . . .in high temporal and spatial detail. . . (replace great)
Done. P3-L1
P3-L13-16: Regarding ablation stakes, I am curious why you did not also use these 17 ablation stake observations as validation for ETI_dist, in the same way you use the AWS as a point observation. Perhaps you did not have stake measurements on the days of the UAV survey. . . However, I am perplexed by the reporting of an RMSE for the stake observations, perhaps you mean the RMSE of the modelled melt (from DSM differencing) at the stake locations? It is important to be clear that your DSM differencing is not communicated "measured" melt, but rather modeled, or estimated, melt using the geodetic method.
We have tried to clarify the source of this reported uncertainty. P3-L26-29
P3-L23: Suggest including the manufacturers name for the SR50.
Done. P3-L25
P3-L31-32: Please provide some justification for why a 5-hour rolling average was chosen, and how samples qualified as being "significantly different" from the population.
We have added further description of this processing. P6-L1-11
P4-L3-4: When referring to resolutions (0.10 m, 0.02 m) specify whether these are horizontal resolution (your cell size) or vertical resolution; also be specific with the reported uncertainty – which is presumably vertical uncertainty? +/-?
Done. P6-L15-17
P6-L13: ". . .temperature was assumed to remain constant across the area." Out of curiosity, do you have a rough idea of what the temperature gradients are in this area? A quick correlation analysis between model error and surface elevations from your last DSM would be a simple way to verify that this assumption is reasonable.
The model now allows temperature to vary, see major revision comment 1.
P6-L15: What do you mean by "modify"? I would suggest giving much more attention to describing how the slope and aspect correction is applied in ETI_dist, particularly given the correlation of your error to aspect later on (see general comment #7)!
P6-L19-30: This is the section where it would be helpful to have some more detail from the previous studies you draw upon build your model and conduct corrections. This section really describes the heart of your model, and any additional detail you can provide will help readers understand the reasoning for your model design and why it performs the way it does. I might even suggest creating a simple flow-chart that illustrates the model inputs and their sources.

We have not included a flow chart, but have tried to expand on the description of both models.

P6-L27: Is there a reason your reported range in albedo goes from high to low? (Rather than "0.1-0.55"?)

New text here. P8-L29

P7-L12: As noted in the general comments, a brief explanation of the Höhle and Höhle paper's approach would be helpful – I am personally curious why the median is used instead of the mean!

A fuller description of the reasoning behind our choice of statistics has been added. P9-10

P7-L19: Why was 1500 cells chosen as the threshold for number of cells contributing to a water flow feature, and can you express this value in the area equivalent (e.g. square meters)?

Added. P10-L19

P8-L8: Uncertainties should be included the modelled and measured values reported here.

Added. P11-L14-17

P9-L4: Be clear what this correlation value is (Pearson correlation coefficient) or by using r = 0.34. Putting the correlation value at the end of this sentence seems awkward, maybe try rewording?

This section has changed. P12-13

P9-L7: Similar to above, correct to include ". . . much lower (r < 0.1, and p > ___). . ."

See above.

P10-L1: Suggest observation rather than "experience" P10-L3: "-0.048 m"; also reword for clarity and include r =.

Done. P14-L13

P10-L15: ". . .deviation of 0.00083 m h-1, which is similar to other. . ."

This has been rewritten. P13-L25

P10-L20: Be specific, which year?

Rewritten for clarity. P14-L1

P10-L32: Specify, horizontal or vertical resolution

Done. P14-L12

P11-L4: "offsets"

Done. P14-L16

P11-L12: "Correlation between aspect. . ." This sentence feels out of place in a description of work by Bash et al. (2018), perhaps try rewording or open this paragraph with this sentence or something similar. E.g. "The correlation (r = X) between aspect and model error . . ."

This paragraph has been removed.

P11-L18: Tighten up wording, e.g. "Bash et al. (2018) measured higher melt rates in active supraglacial streams than on the surrounding ice."

Changed. P14-L31

P11-L23: ". . . between the density of linear. . ."

Rewritten. P15-L1

P12-L9-16: It is unclear what this paragraph contributes to the paper here, rather it seems to interrupt a discussion of water flow features and their impact on melt. Perhaps add more context, otherwise remove.

This has been removed.

P12-L20: ". . .have a greater relative importance."

This has been removed.

P12-L31: What do you mean by "simplifications"?

Clarified. P16-L15

P13-L5: Can you be specific about how you actually build upon the work of Rippin et al. (2015) to estimate albedo? Is this by introducing a scaling approach?

Reworded. P16-L24-25

P13-L10: Unclear what this first sentence is saying regarding "other implementations"

Clarified. P16-L29

P13-L19: It is not clear how Stevens et al. (2018) and the development of weather crusts relates to this study. Either take more time to explain the relevance (in the discussion section) or remove from conclusions.

More explanation has been added in the discussion.

Figure comments

Figure 2. Extending the E and F y-axis down to zero would help your arguments in the text. Check consistency with bold-type for your graph subsets and parenthesis e.g. A) vs (B), and correct formatting of Iin in the 2nd last line. Can you also explain the gap at the beginning of (F)?

Figure is substantially changed.

Figure 4. Recommend including the study dates – ". . . across the study area between July 21 (hh:mm) and July 23 (hh:mm)."

Done.

[revised manuscript text omitted]

---

## Author Response (AR2)

Dear Dr. Farinotti,

We are pleased to submit the suggested revisions and look forward to hearing from you regarding the next steps for publication. Below you will find our response to the reviewer comments and a highlighted version of the revised manuscript.

In addition to the review comments, we felt that the title no longer reflected the content of the paper, after the addition of the energy balance modelling. While there is still focus on the ETI model, the paper looks at the performance of the EB model as well. We have updated the title to better reflect the new contents.

1. I agree with the conclusion on large local variability in melt rates on glacier surfaces, and yet... when one spends time on a glacier in late summer, in both Arctic and temperate regions, the surface is not typically that variable. Stream channels certainly induce incision and additional melting, below the 'reference surface' of a glacier, but the reference surface is relatively even. Certainly not ~2:1 variations in cumulative summer melt. I wonder if the authors could think about and comment on this in the final conclusions, when it comes to heterogeneity of melt.

This is a good point and we have added this to the discussion on P14L26.

2. I like this result of the effects of meltwater features on additional ablation. e.g. p.1, l.14 and in the discussion, but I am wondering if it is possible to separate melting due to streamflow from that due to albedo effects. Water will absorb and also transmit more solar radiation to the ice (the channel bed), giving extra solar and possible thermal melting, if the water warms in the sun. This could contribute to the stream-associated ablation, so it is probably best to mention this as well as the kinetic/mechanical dissipation that is proposed here. Without specific measurements and modelling of these processes, I would recommend just to mention the different possible explanations and not pick a favorite.

We have added a reference to potential for albedo effects on P14L11. We feel that in other places of both the conclusions and discussion the language we use reflects the multiple possibilities for energy transfer, rather than a favorite and we have left that as is (P14L10, P15L21)

3. p.12, This is a good analysis - it shows pretty convincingly that the roughness alone can't explain the discrepancy very well, although part of it.

I worry that much of the systematic underestimation of melt in the distributed models may come from the systematic underestimation of SWin. Is it possible to add a short sensitivity test of this? I would suggest using the observed (AWS) SWin, maybe just adjusting for slope and aspect effects across your study area, vs. the full modelled SW radiation which has significant biases. This could help to say how much of the AWS vs. distributed model results can be explained from this. As it sits now, it looks like the reference AWS site has ~15% more melt energy than the average grid cell in your study area, which is possible but could also just be a reflection of the SW radiation error.

We agree that some of the systematic underestimation is resulting from lower modelled radiation and we have acknowledged this more explicitly in text (P10L16). However, the

discrepancy between measured and modelled melt is greater across the grid than at the AWS site, indicating the difference is not due to solar radiation alone. We do not feel that a sensitivity analysis would add significantly to the results, but rather might show that improved radiation might reduce the systematic underestimation. We have acknowledged the shortcomings of the radiation model in text (P7L7,P10,L15) and feel this makes clear the potential improvements in this (as with any) model.

Specific comments

p.1, l.10, it would be useful to also report the total melt over the study period, and/or to report the model confidence as a percentage of the total melt. i.e. the models agree within *% of the observations.
Done P1L9

p.3, l.19, uncertainty reported for the melt rate - is this over 3 days? - suggest reporting a per-day rate, for better comparison with the numbers below (0.06 m/d)
We feel that reporting this uncertainty as a daily rate is misleading as it is linked closely to image resolution, reporting it as a rate would make it appear artificially low.

p.3, l.21, there should be only 1 value for an average, Do you mean the range of daily melt rates? Or would it be better to report 0.042 +/- 0.012 m/d?
We have changed this to be clearer P3L21

p.3, l.27 and p.4, l.2, suggest using "surface height" rather than "position" for the SR50 signal. Just as you are not looking at (x,y) here, which I think is implied by "position"
Done P3L27, P4L1

p.5, l.22, For clarity, do you mean MJ per 12 hour period, MJ per day, or MJ over your full period?
Clarified P5L22

p.6, l2, suggest \sigma for Stefan-Boltzmann's constant, to be conformist.
We recognize that \sigma is the standard notation for Stefan-Boltzmann's constant, but have chosen \delta to avoid confusion with standard deviation, for which \sigma is also the standard notation.

p.10, l.20, Can you remind us here about the values of the observed melt at the AWS site, from a) the SR50 and b) the UAV results? This would help with the evaluation.
This has been added P10L23

p.10, l.24, "all four models underestimate melt at the point" - but aren't they tuned at the AWS site, and with a mean 0 residual as noted above? Do you mean underestimate over the study area?

We found a calculation error in our total residual statistics which has been corrected here and in Table 2, given that we have clarified this sentence (P10L26). However, we also note that although models are tuned at the AWS they will never perfectly predict observations, and the text here is indeed referring to tuned models at the measurement site.

p.12, l.12, also there is a strong diurnal variation in the zenith angle - places with 24-hour light still have strong diurnal cycles, e.g. greater than 2:1 variations in incoming SW radiation at local noon vs. local midnight, depending on the latitude
We clarified this and added an explicit reference to zenith angle variations, which we did not mean to ignore previously P12L14

p.12, l.13, "in. The strength of that variation is much greater in the EB models than the ETI models". This does not make sense to me - don't both models use incoming SW, using the same model for this? As this reads, the shortwave radiation varies more for the EB model than the ETI model. Do you mean the errors or residuals, rather than the SW radiation here?
We have clarified this P12L16

p13, l.5, here, it is both EB and ETI models that come up short, non? Maybe refer to them as the "distributed melt models" ?
As this discussed is regarding the TF coefficient, it does not apply to EB models and we have left this unchanged.

Figure 2 caption. For the SW, it would be helpful to note the duration of time used for the units here MJ/m2) as this is a cumulative value - are these hourly values? In which case it is cumulative or total hourly (vs. hourly averaged) SW. It is more like PDD with MJ.
The caption already states that all values are hourly averages.

Figure 4b - is there a reason to show just the second half of July here? it works fine, maybe just not clear why some things are all of July and some just the second half.
We have explicitly noted the SR50 measurements are only available July 14-31 in text (P3L26) and in this caption.

[revised manuscript text omitted]

---

## Author Response (AR3)

Dear Dr. Farinotti,

We are very pleased to submit a revised manuscript including the technical corrections you suggested. We have updated the title to: "Surface melt and the importance of water flow – an analysis based on high-resolution UAV data for an Arctic glacier"

On page 15, line 26, we have added "…including the potential role of the lower water albedo when compared to ice." To more clearly address the reviewer's comments.

On page 12, line 9, we have added two sentences regarding the role of lower solar radiation in the distributed model: "Lower modelled melt across the study area can be partly explained by lower modelled radiation, which was noted at the AWS site. However, the discrepancy between measured and modelled melt is greater across the grid than at the AWS site, indicating the difference is not due to solar radiation alone."

P10 L16: "for the study period" ("the" is missing)

Done.

Table 2: Please use the table's caption to define "Residual \sigma". I assume it is the standard deviation of the residuals as defined at P9 L10?

We changed the notation slightly to help clarify what this means and added a description in the caption.

We look forward to hearing from you again and the paper being published.

Sincerely,

Eleanor Bash and Brian Moorman

[revised manuscript text omitted]